# Selfing is the safest sex for *Caenorhabditis tropicalis*

Luke M Noble[1,2]*, John Yuen[1], Lewis Stevens[3], Nicolas Moya[3], Riaad Persaud[1], Marc Moscatelli[1], Jacqueline L Jackson[1], Gaotian Zhang[3], Rojin Chitrakar[4], L Ryan Baugh[4], Christian Braendle[5], Erik C Andersen[3], Hannah S Seidel[6]*, Matthew V Rockman[1]*

[1]Department of Biology and Center for Genomics & Systems Biology, New York University, New York, United States; [2]Institute de Biologie, École Normale Supérieure, CNRS, Inserm, Paris, France; [3]Department of Molecular Biosciences, Northwestern University, Evanston, United States; [4]Department of Biology, Duke University, Durham, United States; [5]Institut de Biologie Valrose, Université Côte d'Azur, CNRS, Inserm, Nice, France; [6]Department of Biology, Eastern Michigan University, Ypsilanti, United States

**Abstract** Mating systems have profound effects on genetic diversity and compatibility. The convergent evolution of self-fertilization in three *Caenorhabditis* species provides a powerful lens to examine causes and consequences of mating system transitions. Among the selfers, *Caenorhabditis tropicalis* is the least genetically diverse and most afflicted by outbreeding depression. We generated a chromosomal-scale genome for *C. tropicalis* and surveyed global diversity. Population structure is very strong, and islands of extreme divergence punctuate a genomic background that is highly homogeneous around the globe. Outbreeding depression in the laboratory is caused largely by multiple Medea-like elements, genetically consistent with maternal toxin/zygotic antidote systems. Loci with Medea activity harbor novel and duplicated genes, and their activity is modified by mito-nuclear background. Segregating Medea elements dramatically reduce fitness, and simulations show that selfing limits their spread. Frequent selfing in *C. tropicalis* may therefore be a strategy to avoid Medea-mediated outbreeding depression.

**\*For correspondence:**
noble@biologie.ens.fr (LMN);
hseidel@emich.edu (HSS);
mrockman@nyu.edu (MVR)

**Competing interests:** The authors declare that no competing interests exist.

## Introduction

Sex and outcrossing are common, but costly, and taxa have repeatedly evolved mating systems to avoid them. Such transitions set the advantages of selfing, such as reproductive assurance, against long-term adaptability (*Otto, 2009*). Selfing has profound consequences for evolution due to changes in effective recombination, homozygosity, and migration, leading to a net reduction in effective population size. Mixed mating systems, combining some form of selfing with occasional outcrossing, are a frequent compromise (*Chelo et al., 2019*; *Cutter, 2019*; *Escobar et al., 2011*; *Goodwillie et al., 2005*; *Igic and Kohn, 2006*; *Jarne and Auld, 2006*).

Variation in mating systems is especially familiar in plants, but has also been one of the longstanding attractions of nematode biology (*Nigon and Félix, 2017*). Just within Rhabditidae, this aspect of life history now spans systems with separate males and females (gonochorism), males and self-fertile hermaphrodites (androdioecy; *Kanzaki et al., 2017*; *Mayer et al., 2007*), males, females, and hermaphrodites (trioecy; *Chaudhuri et al., 2015*; *Kanzaki et al., 2017*), asexual reproduction where sperm does not contribute genetic material (parthenogenesis, gynogenesis; *Fradin et al., 2017*; *Grosmaire et al., 2019*), and alternating generations of hermaphroditism and dioecy (heterogony; *Kiontke, 2005*). Within the *Caenorhabditis genus*, the androdioecious system of males and self-fertilizing hermaphrodites has evolved three times independently (*Ellis, 2017*). Hermaphrodites are

morphologically female, but during larval development they generate and store sperm for use as adults. Hermaphrodites cannot mate with one another, and in the absence of males all reproduction is by self-fertilization.

*Caenorhabditis tropicalis* was identified by Marie-Anne Félix in 2008 (*Kiontke et al., 2011*), and investigations of its reproductive biology have shed light on the mechanistic basis of transitions to selfing (*Wei et al., 2014*; *Zhao et al., 2018*). However, a comprehensive reference genome for the species is lacking and relatively little is known of its biology and ecology. Global sampling indicates a more restricted range than that of the other selfers (*Félix, 2020*), and the single study of *C. tropicalis* population genetics and reproductive compatibility found extremely low levels of genetic diversity at a handful of loci (*Gimond et al., 2013*). Crosses among, and sometimes within, locales revealed outbreeding depression, a result common to the selfers (*Baird and Stonesifer, 2012*; *Dolgin et al., 2007*; *Ross et al., 2011*) but in stark contrast to gonochoristic species, where estimated diversity is often orders of magnitude higher and inbreeding depression can be severe (*Barrière et al., 2009*; *Dolgin et al., 2007*; *Gimond et al., 2013*). Outbreeding depression was particularly acute in *C. tropicalis*, with embryonic lethality and developmentally abnormal $F_2$ progeny common among certain hybrid crosses (*Gimond et al., 2013*). Male mating ability was also found to be generally poor, though highly variable, which together with low genetic diversity suggests an especially high rate of selfing.

Theoretical explanations for the evolution of selfing consider a balance between benefits and costs. Classical models start with the doubling of reproductive rate achieved by selfers (through the elimination of males in the case of *Caenorhabditis*) and note that selfing will evolve when the reduction in reproduction due to inbreeding depression is less than one half (*Cutter, 2019*; *Goodwillie et al., 2005*; *Lande and Schemske, 1985*; *Lively and Lloyd, 1990*). Subsequent work has added a variety of factors on both sides of the balance. Selfing has benefits for reproductive assurance (*Baker, 1955*; *Theologidis et al., 2014*), allowing a single individual to colonize a new environment, and it allows for rapid adaptation when traits are jointly determined by maternal and zygotic genotypes (*Drown and Wade, 2014*). On the other hand, selfing slows adaptation to new environments or pathogens by reducing recombination and genetic diversity, while increasing the load of weakly deleterious mutations (*Cutter, 2019*; *Kamran-Disfani and Agrawal, 2014*; *Morran et al., 2009*, *Morran et al., 2011*). Many features of the selective environment (e.g., stabilizing vs. directional) and the genetic architecture of fitness (e.g., recessive deleterious variation vs. overdominance as the cause of inbreeding depression) shape the balance of factors, and vary among species and populations (*Goodwillie et al., 2005*). The ledger of relevant factors runs long, but as we show, it may still be incomplete.

To study the causes and effects of mating system evolution in Caenorhabditis, we assembled a chromosome-scale genome for *C. tropicalis*, oriented by linkage data from recombinant inbred lines (RILs). We show with short-read mapping against this reference that the population structure of a global sample of isolates is very strong, which is an expected side-effect of selfing. We also find extreme heterogeneity in the distribution of genetic diversity across the genome. Finally, we investigate the causes of strong outbreeding depression, another expected side-effect of selfing, in a cross between divergent isolates. Widespread outbreeding depression observed in the three species of selfing Caenorhabditis has been interpreted as evidence for the well-understood process of Dobzhansky–Muller epistasis. Here, we show that it is due largely to a different process in *C. tropicalis*: maternal-effect haplotypes that kill offspring that do not inherit them. Haplotypes that behave in this way are known as Medea elements (*Beeman et al., 1992*; *Beeman and Friesen, 1999*), and they represent a form of post-zygotic gene drive (*Price et al., 2020*; *Wade and Beeman, 1994*). Medeas impose severe fitness costs on heterozygous mothers, and we hypothesize that these elements select for a high selfing rate in *C. tropicalis*, by adding another factor to the balance on the side of selfing. In other words, frequent selfing may be a consequence of outbreeding depression as much as its cause.

## Results

### Heritable variation in outcrossing rates

To better understand variation in outcrossing in *C. tropicalis*, we tested a global sample of five isolates for their propensity or ability to outcross. For each of these strains, from Hawaii, Panama, French Guiana, Cape Verde, and Réunion Island, we assayed the probability that an individual would produce cross-progeny when paired with a single individual of the other sex. Mating success was scored as a binary trait, with crosses scored as successfully mated if multiple male offspring were observed on the plate. This scoring system allowed for the possibility of rare male offspring to be produced by X nondisjunction in hermaphrodites, which typically occurs at a low rate (~1% or lower, see below). All factorial crosses produced progeny, but strains varied significantly in their propensity to outcross, both as males and as hermaphrodites (*Figure 1A*). We also observed interaction effects, where the average crossing probability of male and hermaphrodite strains was not predictive of the success of the strains in combination. At one extreme, the pairing of JU1373 hermaphrodites from Réunion with JU1630 males from Cape Verde yielded no cross-progeny from 22 trials, while at the other extreme, Hawaiian QG131 hermaphrodites and South American NIC58 males yielded cross-progeny in each of 34 trials. Male crossing probability was much more variable than that of hermaphrodites, consistent with relaxed selection (residual deviance of 179.2 vs. 85.2, null deviance 232.6, binomial linear model) (*Cutter, 2019*; *Jalinsky et al., 2020*; *Noble et al., 2015*; *Palopoli et al., 2008*; *van der Kooi and Schwander, 2014*). Same-strain pairings were indistinguishable from inter-strain pairings (p=0.88, likelihood ratio test [LRT] of binomial linear models). This analysis shows that

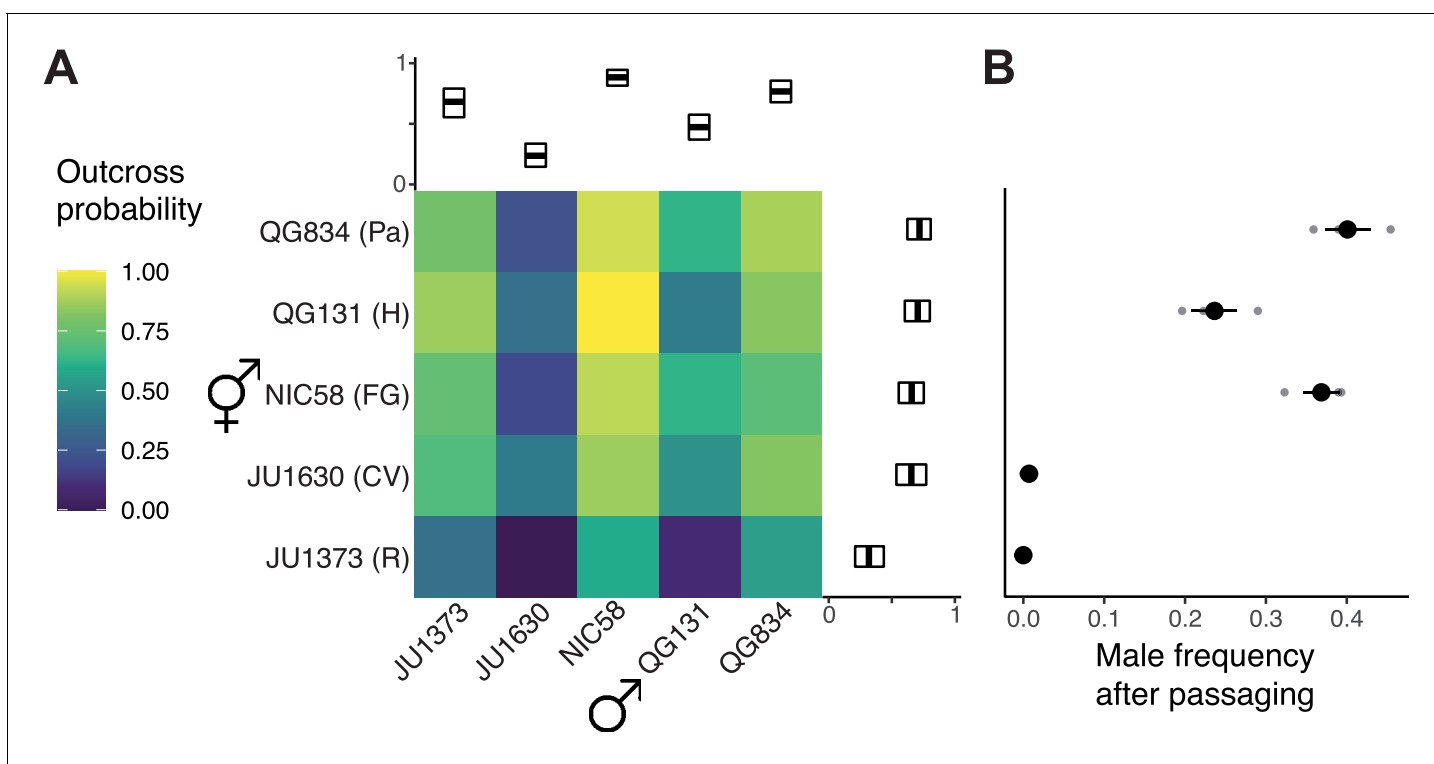

**Figure 1.** Outcrossing dynamics. (**A**) Outcrossing probability in reciprocal crosses. Mating success was scored as a binary trait in 22–34 trials (biological replicates) per cross. Marginal means with bootstrap 99% confidence intervals are shown. (**B**) Strains vary in their male frequency after 10 generations of passaging at large population size (mean and standard error of three biological replicates). R: Réunion Island, CV: Cape Verde, FG: French Guiana, H: Hawaii, Pa: Panama. Data are in *Figure 1—source datas 1* and *2*.

The online version of this article includes the following source data for figure 1:

**Source data 1.** outcrossProbability.tsv.zip; wild isolate outcross probability trials.
**Source data 2.** malePassaging.tsv.zip; wild isolate passaging male frequency.

wild isolates of *C. tropicalis*, and males in particular, vary greatly in their propensity or ability to outcross.

Given extensive variation in outcrossing and the independence of male and hermaphrodite components of this trait, we asked whether strains could maintain males over time, or whether hermaphrodite selfing would drive them from populations (*Stewart and Phillips, 2002*). We founded single-strain populations with three hermaphrodites and five males, allowed them to expand, and then serially transferred their descendants for 10 generations under standard lab conditions. This experiment tests additional components of fitness, such as fecundity, in a continuously expanding population. At the end, two strains, JU1630 and JU1373, had lost males and were reproducing solely by selfing; these are the strains with the lowest cross success in our single-worm pairings (*Figure 1B*). The other strains retained males at frequencies of 20–40%. This dichotomy resembles that seen among *C. elegans* strains, where some strains become exclusive selfers, while others can maintain high rates of outcrossing in simple environments (*Stewart and Phillips, 2002*; *Teotónio et al., 2006*; *Teotonio et al., 2012*). We chose the strain with the most-outcrossing males, NIC58 from French Guiana, and the strain with the least-outcrossing hermaphrodites, JU1373 from Réunion Island, as focal strains for an investigation of the genetics and population biology of *C. tropicalis*.

## A chromosomal genome for NIC58

To generate a reference genome for *C. tropicalis* we used deep PacBio long-read sequencing of NIC58, and then applied genetic linkage data from RILs to assess and orient multiple assemblies. RILs were derived from a cross between a JU1373 hermaphrodite and a NIC58 male, by selfing $F_2$ hermaphrodites from a single $F_1$ for 10 generations. We genotyped lines by shotgun sequencing, called diallelic single nucleotide variants (SNVs), and inferred parental ancestry by Hidden Markov Model, using data for 119 RILs for genetic map estimation. During RIL construction, 12.2% of lines did not survive, consistent with outbreeding depression. The genotypes of the surviving RILs revealed strong transmission ratio distortion favoring JU1373 alleles at two loci on chromosomes III and V (Figure 5), which we return to later.

Using the RIL recombination data to evaluate assemblies, we selected one of five fully concordant with the genetic data. We then closed gaps of estimated 0 cM distance, using junction-spanning sequences in other assemblies, followed by local long-read mapping. Five gaps of greater than 0 cM remain, which will require longer reads to resolve. The resulting nuclear genome assembly comprised 81.3 Mb in 15 sequences. The X chromosome assembled as a single contig, and all other chromosomes were oriented genetically into pseudochromosomes. We assembled a 13,935 bp mitochondrial genome from short-reads, followed by circular extension with long-reads. To better assess genetic variation between the RIL founder strains, we also assembled draft nuclear (81 Mb span, 4.2 Mb NG50) and mitochondrial (13,911 bp) genomes for JU1373 from similar read data (see Materials and methods; we did not attempt to bring this assembly to pseudochromosomes here). We annotated nuclear genomes using mixed-stage short-read RNAseq data, calling 21,210 protein-coding genes for NIC58 and 20,829 for JU1373, and we annotated the mitochondrial genomes by homology. Thus, this pipeline provided a high-quality assembly for NIC58 and a highly contiguous draft genome for JU1373.

## Surveying genetic diversity worldwide

*C. tropicalis* is widely distributed within 25° of the Equator and absent outside this region (*Félix, 2020*). To begin a global survey of genetic diversity and population structure, we sequenced an additional 22 isolates that broadly represent the species' global range (Figure 3A). The collection spans Africa, Asia, and South and Central America, but large equatorial regions, notably in Central Africa and Southeast Asia, are not yet represented. Our sample included 16 American isolates (eight from the Caribbean, eight from Central and South America), four from East Asia, three from Africa, and one from the Central Pacific (*Figure 3—source data 1*). We called variants against the NIC58 reference genome from short-read mapping, hard-filtered to 794,676 diallelic SNVs on the nuclear genome (genotype set 1, *Supplementary file 4*; see Materials and methods), and selected 397,515 sites with fully homozygous calls and no missing data (genotype set 2, *Supplementary file 5*) for exploration of population structure. We called mitochondrial variants separately, and filtered similarly, retaining 166 (of 197 hard-filtered) SNVs.

Previously *Gimond et al., 2013* sequenced 5.9 kb across nine nuclear, protein-coding loci in 54 isolates (mostly from French Guiana in South America, but including African isolates from Cape Verde and Réunion Island, and our Pacific isolate from Hawaii) and found nine SNVs, equating to a per-site Watterson's θ around 0.00034. Though not directly comparable, the genome-wide estimate of nucleotide diversity provided here is around three times higher (genotype set 1, median value across 20 kb windows = 0.00097), with mitochondrial diversity higher again as expected (0.0038). These values likely underestimate species-wide variation because of short-read mapping bias – we adjust for missing data, but missing data may not, in fact, be missing from genomes. Indeed, we found rampant missingness in our data; up to 1.3% of the NIC58-alignable fraction of the genome lacks aligned reads among any single isolate, and 7.8% of the alignable genome lacks reads in at least one (considering only those with >25 × mapped and paired reads, for which missing data and sequencing depth are uncorrelated, $r = 0.13$, p=0.65). These results establish that *C. tropicalis* shows very low genetic diversity genome-wide, but also suggest this homogeneity is broken by many regions sufficiently divergent that reference-based mapping fails.

## *C. tropicalis* genetic diversity is highly heterogeneous along chromosomes

Selfing has complex effects on population dynamics and genome evolution. A general expectation is a reduction in effective population size $N_e$ proportional to the frequency of selfing, by a factor of up to two, and the strength of background selection, by potentially much more (*Charlesworth, 2012*; *Nordborg and Donnelly, 1997*). The strength of selection acting on genetic variation is proportional to the product of $N_e$ and the selection coefficient *s*. Selfing is therefore expected to both lower genetic diversity, making evolution more reliant on new mutations, and raise the threshold below which mutations are effectively neutral. We found that the distribution of SNV diversity along all chromosomes is extremely heterogeneous in *C. tropicalis*. Recombination is relatively homogenous within the large recombination rate domains, at least in *C. elegans* (*Bernstein and Rockman, 2016*; *Kaur and Rockman, 2014*), and is therefore not expected to generate heterogeneity in the effects of linked selection at these small scales. Background diversity in *C. tropicalis* (median $\theta_w$) is the lowest among the three selfers, and the median number of SNV differences between isolates (π) in 10 kb windows is just 3.2 on chromosome centers, and less than double that on arms (genotype set 2). Variance around the background is almost eight times that of *C. elegans* and more than 100 times that of *C. briggsae* (data in *Figure 2A*). We used kernel density smoothing of the binned distribution of $\theta_w$ to partition the genome into segments of very high diversity (the long right tail of divergent outlier regions) and segments with background levels of diversity (e.g., see *Figure 2A*). Heterogeneity is often highly localized: at 10 kb scale, 141 divergent peaks fall to background within 30 kb or less (see Materials and methods), and divergent regions cover just over 14% of the NIC58 genome in sum. Genetic diversity in *C. tropicalis* is thus typified by a near-invariant background, suggesting very recent global shared ancestry, punctuated by regions of high divergence, consistent with a possible role for balancing selection.

Finally, to gain a view of *C. tropicalis* genetic diversity less subject to reference-mapping bias, we aligned the draft JU1373 genome against NIC58, calling variants and assessing copy number variation from alignment depth (*Li, 2018*). From 78.86 Mb of aligned bases (69.18 at single copy), we saw a 37% increase in SNVs over reference-based mapping, and a sum of 1.23 Mb in insertion-deletion variation including 388 variants of length greater than 1 kb. SNV divergence in 10 kb windows commonly exceeded 10% on the arms (*Figure 2B*), and total divergence (the sum of variant length differences relative to NIC58) exceeded 30% in windows on every chromosome. Reference-based SNV calling thus dramatically underestimated the true levels of genetic diversity at divergent loci, which were comparable to current estimates for gonochoristic species and to analogous patterns recently described in *C. elegans* and *C. briggsae* (*Lee et al., 2020*). In the face of such high divergence, at single base and multigene-scale, long-read genomes and variant graph genome representation may be required to more fully describe species-wide variation (*Garrison et al., 2018*). Deeper population genetic data may also allow inference of divergent foci within these loci.

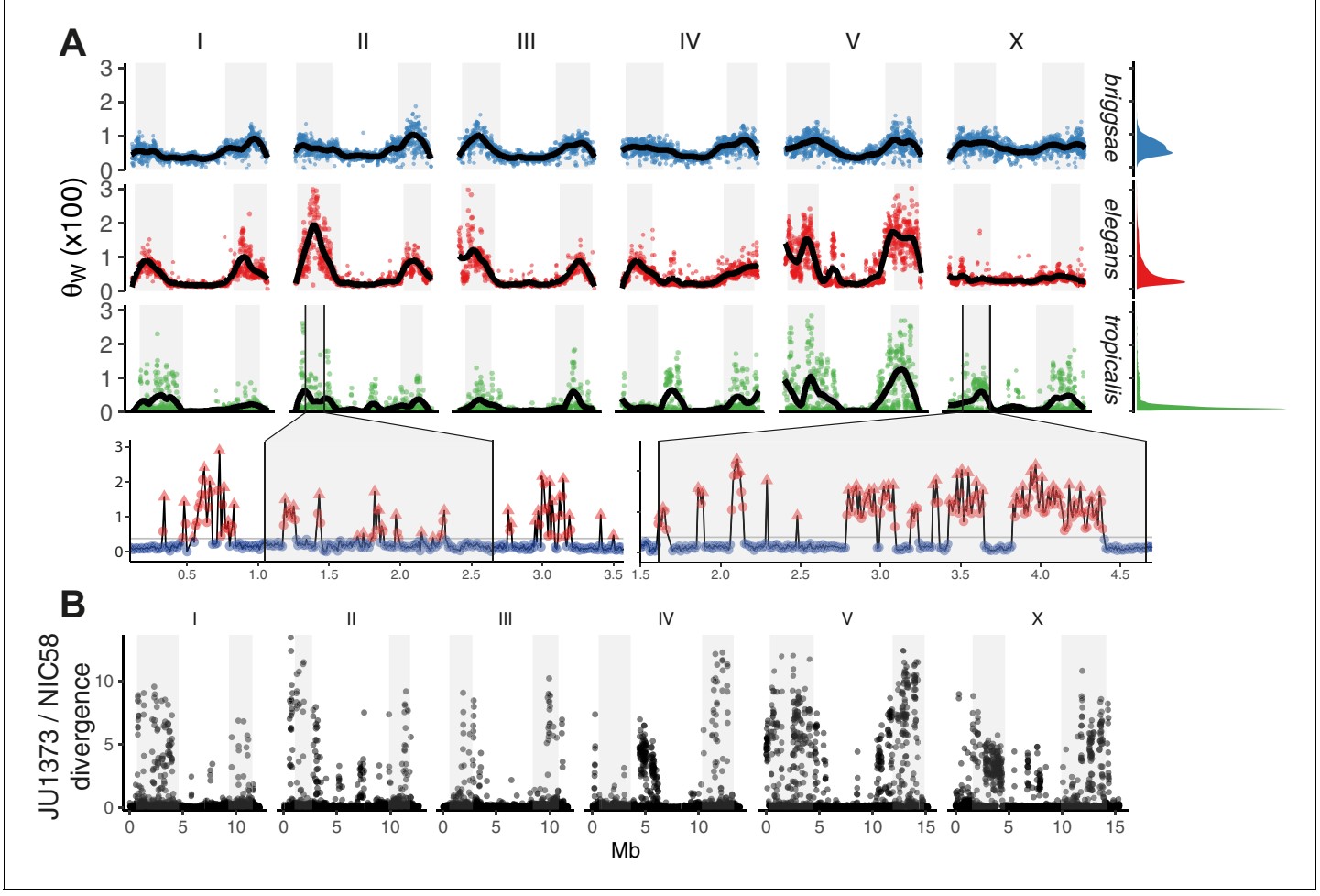

**Figure 2.** The distribution of genomic diversity in selfing *Caenorhabditis* species. (**A**) Nucleotide diversity across chromosomes based on 24 strains for *C. tropicalis*, 35 strains for *C. briggsae*, and 330 isotypes for *C. elegans* (Watterson's θ [× 100] in non-overlapping 20 kb windows, physical distance normalized across species and chromosomes). The denominator in Watterson's estimator uses the mean number of strains with non-missing calls per window rounded to the nearest integer. 12 outliers for *C. elegans* are outside the plotted range. Differences in heterogeneity across species are apparent from the marginal density plots, and from dispersion around the locally weighted polynomial (LOESS) fit to the data in black. Levels of variation at loci in *C. tropicalis* centers approach those of arms for chromosomes II, IV, V, and X. Arm recombination rate domains are shaded, and regions on the left arms of chromosome II and the X are magnified below. Here, triangles are local peaks called at 10 kb scale by segmenting divergent regions (red) from background (blue) at the threshold indicated by a gray line (see Materials and methods), the y-axis is as in the main plot, and the x-axis is unnormalized physical distance (Mb). Data are in *Figure 2—source data 1*. (**B**) Genetic diversity between JU1373 and NIC58 from genome alignment, shown as single nucleotide variant (SNV) differences (1 - % identity) in 10 kb non-overlapping windows. Data are in *Figure 2—source data 2*.

The online version of this article includes the following source data for figure 2:

**Source data 1.** selfer_theta_20 kb.tsv.zip; Binned nucleotide diversity for *C. elegans*, *C. briggsae*, and *C. tropicalis*.
**Source data 2.** JU1373-NIC58.alignmentCoverage.tsv.zip; JU1373 and NIC58 identity and copy number variation (Minimap2 alignment).

## *C. tropicalis* shows strong continental population structure with local heterogeneity

To examine population structure worldwide, we decomposed NIC58-reference-based genetic relatedness of the 24 isolates into its principal components. We observed strong structure, with the top axis differentiating three African isolates from all others and accounting for almost 75% of nuclear genetic relatedness (*Figure 3B*). The close clustering of the three African lines, isolated across a transect spanning more than 9000 km from the Atlantic island of Cape Verde, off the Westernmost coast of continental Africa, to Réunion Island in the Indian Ocean, is remarkable given the large

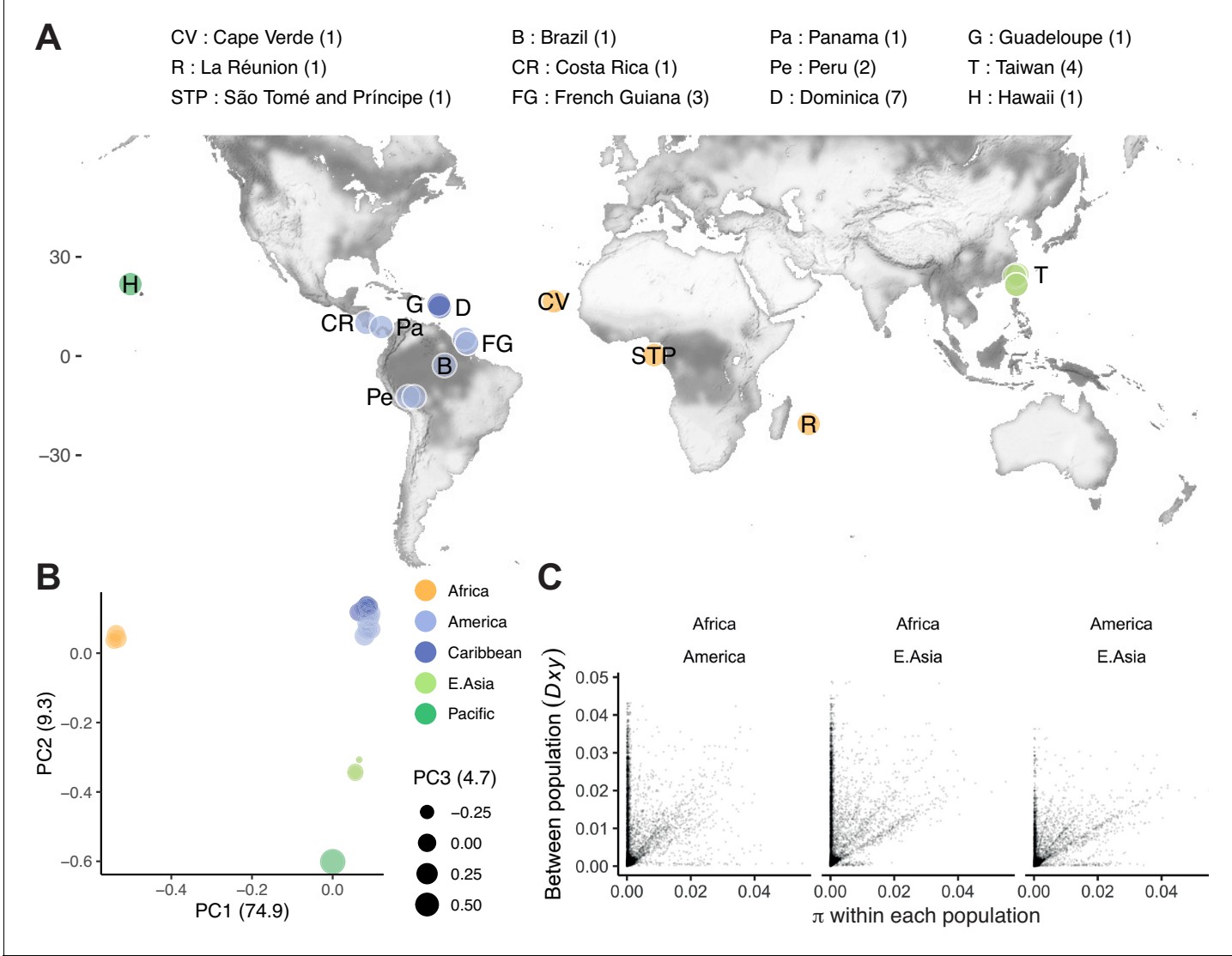

**Figure 3.** Population structure. (**A**) The distribution of 24 isolates (numbers per locale are shown in the legend; Data in *Figure 3—source data 1*), colored by groupings in (**B**), where principal component analysis of nuclear genomic similarity identifies largely discrete populations. (**C**) Genetic diversity is found mostly, but not entirely, within populations. For three populations with at least two lines (<99% single nucleotide variant [SNV] identity), within population diversity (π) is plotted for each population against between population diversity (*Dxy*; *Nei and Li, 1979*; 10 kb scale). Data in B and C are based on *Supplementary file 4*.

The online version of this article includes the following source data and figure supplement(s) for figure 3:

**Source data 1.** isolateMetadata.tsv.zip; metadata for *C. tropicalis* wild isolates.
**Figure supplement 1.** Chromosome and mitochondrial population structure.
**Figure supplement 2.** The average number of single nucleotide variant (SNV) differences among all pairwise comparisons within (π; upper, with the global value for all pooled samples plotted in blue across each panel) and between (*Nei and Li, 1979*; lower) populations (10 kb non-overlapping windows, adjusted for the mean fraction of missing data per window).

geographic distances separating these locales, though it is consistent with the occurrence of globally distributed haplotypes in *C. elegans* (*Lee et al., 2020*). The second principal component differentiates Western Pacific samples (Hawaii, Taiwan) from all others, and the third principal component largely differentiates two of four Taiwanese samples from Hawaii. These three dimensions account for 89% of the genetic variance, which is of similar magnitude to the variance explained by the first three principal components in *C. briggsae* (based on 449,216 SNVs with no missing data among 34 lines).

The genome-wide view masks heterogeneity at the chromosome level. Notably, chromosome IV shows much more complex patterns of relatedness, and both chromosome V and the mitochondrial genome provide evidence of recent admixture, with strain QG834 from Panama clustering with strains from East Asia and the Pacific. Structuring of the X is particularly extreme, with essentially two haplotypes present in our sample, African and non-African, and this split accounts for 98% of the genetic variance (*Figure 3—figure supplements 1* and *2*).

While the structure revealed by this analysis explains most of the genetic variance, we also observe variance within these populations at divergent regions. A minority of loci are divergent both within and across populations (*Figure 3C*). We also find a handful of highly divergent regions that vary only within populations. For example, loci on chromosomes IV and V vary among seven isolates from a single collection made over 1 month from the small island nation of Dominica. The structure of *C. tropicalis* populations therefore combines strong, global differentiation, as seen between tropical and temperate *C. briggsae* clades; widespread homogeneity, as seen in *C. elegans* outside the Pacific; and diversity at the very local scale, common to all three selfing species (*Andersen et al., 2012*; *Barrière and Félix, 2007*; *Félix et al., 2013*; *Haber et al., 2005*; *Sivasundar and Hey, 2005*).

## Quantitative genetics of outcrossing

As a first step toward understanding the genetic basis of variation in outcrossing, we scored hermaphrodites from the NIC58 × JU1373 RILs for their probability of producing cross offspring in matings with NIC58 males. As expected from their differences in outcrossing rate in single worm and bulk passaging assays (*Figure 1*), we observed considerable variation among lines, including transgressive segregation (*Figure 4A*). Linkage mapping detected a significant effect of a locus on the center of the X chromosome (*Figure 4B*), which explained close to 15% of the variance in hermaphrodite outcrossing probability. Although the difference in equilibrium male frequency between JU1373 and NIC58 is likely mediated by their different outcrossing rates, we also observed differences in the rate of spontaneous male production due to X nondisjunction during hermaphrodite meiosis. The self-progeny of NIC58 hermaphrodites was 0.8% male (21/2580) versus 0.06% in JU1373 (2/3088); given a characteristic brood size of 100–150, these numbers imply that most NIC58 self broods include a male, and most JU1373 self broods do not. These two strains thus differ heritably in male crossing ability, hermaphrodite crossing ability, equilibrium sex ratio, and spontaneous male production rate, providing multiple paths for the evolution of outcrossing rate.

## RIL transmission ratio distortion and excess heterozygosity

The RIL genotypes revealed strong transmission ratio distortion in two genomic regions: the left arm of chromosome III and the right arm of chromosome V (*Figure 5A*). Both were strongly skewed

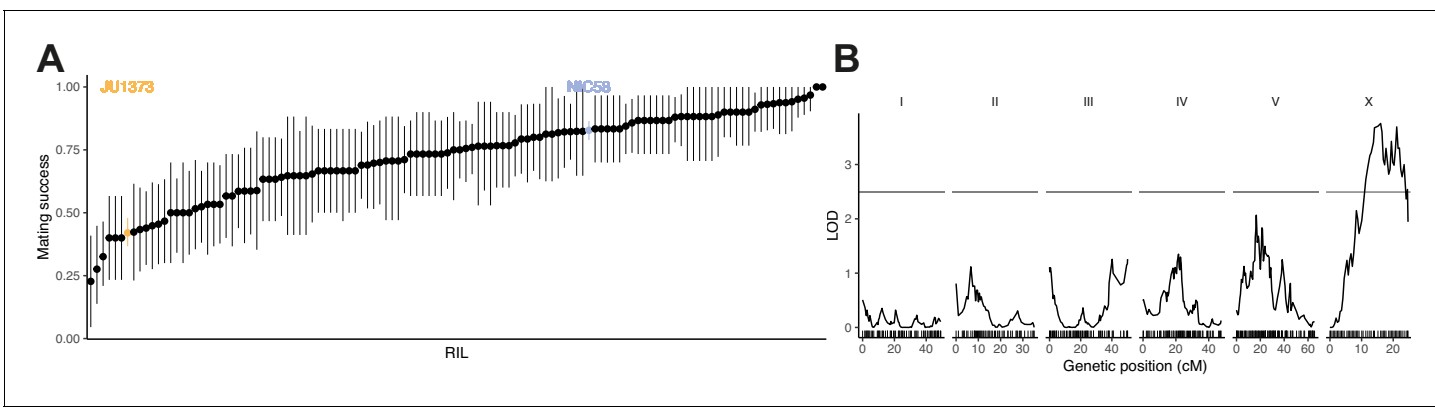

**Figure 4.** Recombinant inbred lines (RILs) vary in their hermaphrodite crossing probability. (**A**) Means and 95% bootstrap confidence intervals from binary trials are shown for RILs and their parents. Data are in *Figure 4—source data 1*. (**B**) Quantitative trait locus mapping for hermaphrodite crossing probability (genome-wide 0.05 significance threshold from 1000 phenotype permutations shown in gray, n = 118 RILs). Data are based on *Supplementary file 1*.

The online version of this article includes the following source data for figure 4:

**Source data 1.** RIL_mating.tsv.zip; RIL outcross probability trials.

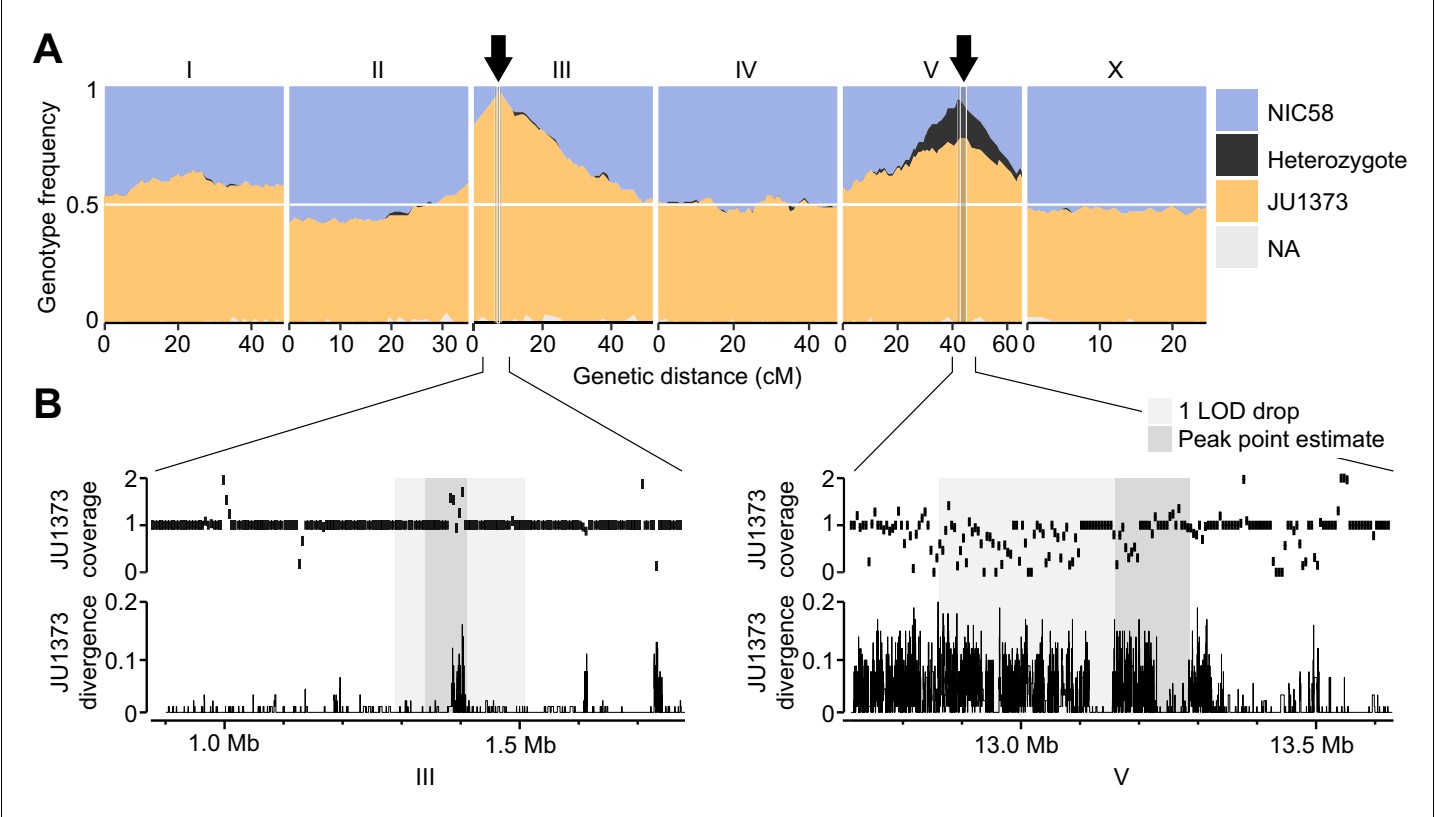

**Figure 5.** Two genomic regions show strong transmission ratio distortion. (**A**) Recombinant inbred line (RIL) genotype frequencies and peaks of transmission ratio distortion (arrows). Shaded areas are 1 LOD drop intervals and peak point estimates. Genome-wide data are based on *Supplementary file 1*, and multilocus transmission ratio distortion genotype tables are in *Figure 5—source data 1*. (**B**) Fold coverage and single nucleotide variant (SNV) divergence of JU1373 relative to NIC58. Fold coverage is in 5 kb windows, divergence is 1-identity in 100 bp windows. Data are based on *Figure 2—source data 2*.

The online version of this article includes the following source data and figure supplement(s) for figure 5:

**Source data 1.** RIL_distortion.tsv.zip; genotype tables at transmission ratio distortion peaks on chromosomes I, III, and V.

**Figure supplement 1.** Models of Medea-mediated gene drive and expected $F_2$ arrest proportions.

toward JU1373 homozygotes, which reached a frequency of 97% on chromosome III and 78% on chromosome V. The chromosome V locus also showed an excess of heterozygotes (16% of RILs) compared to the neutral expectation after 10 generations of selfing ($1/2^{10}$ = 0.1%). RILs that retained heterozygosity on chromosome V also showed a significant enrichment of JU1373 genotypes on chromosome I (18.2–20.7 cM), which itself showed mild distortion in favor of JU1373 (Fisher's exact test, p<0.001, *Figure 5—source data 1*). These data indicate that selection during RIL construction strongly favored JU1373 alleles on chromosomes III and V, with complex selection at the locus on chromosome V favoring heterozygotes over JU1373 homozygotes under some conditions.

## Transmission ratio distortion is not due to simple mitochondrial-nuclear incompatibilities

Strong selection during RIL construction is consistent with previous reports of extensive outbreeding depression (*Gimond et al., 2013*). Yet simple genetic incompatibility between two nuclear loci is not expected to favor one parental allele to the exclusion of the other. Exclusion of one parental allele can occur, however, if an allele from the male parent (NIC58) is incompatible with the mitochondrial genome of the hermaphrodite parent (JU1373). Under this scenario, RILs homozygous for the male parent allele at loci showing transmission ratio distortion should be sub-viable or sub-fertile. We examined RILs of such genotypes and found that their growth characteristics were superficially

normal, with 93.1–98.3% (n = 151–679) of embryos developing into adults with parental developmental timing (Figure 7A). This finding indicated that the cause of transmission ratio distortion was more complex than a simple mitochondrial-nuclear incompatibility between these NIC58 alleles and JU1373 mitochondria.

## Transmission ratio distortion and excess heterozygosity are caused by Medea loci

An alternative explanation for transmission ratio distortion among RILs is that the distorted loci independently exhibit post-zygotic killing dynamics, similar to those seen in *C. elegans* at the *zeel-1/peel-1* and *pha-1/sup-35* loci (*Ben-David et al., 2017*; *Seidel et al., 2008*). At these loci, a maternal- or paternal-effect locus loads a toxin into eggs or sperm that poisons zygotic development; subsequent zygotic expression from the same locus provides an antidote (*Ben-David et al., 2017*; *Seidel et al., 2011*). Independent of the precise mechanism, loci with this pattern of parental-by-zygotic genetics of lethality are classified as Medea-type elements, named (in part) for their maternal-effect dominant embryonic arrest (*Beeman et al., 1992*). We show below that the *C. tropicalis* distorted loci fit this mode of inheritance, and broaden the Medea classification here to encompass post-embryonic arrest. A Medea model predicts that the JU1373 genome is homozygous for two independent Medea haplotypes, on chromosomes III and V, each encoding a hypothetical toxin-antidote pair. (In our analysis of this model we describe it as a toxin-antidote system for convenience, though the underlying mechanism of maternal-effect dominant lethality and zygotic-effect dominant rescue may be different.) Under this model (*Figure 5—figure supplement 1*), all $F_2$ progeny from a NIC58 × JU1373 cross will be exposed to toxins, but only some will inherit antidotes; animals not inheriting both antidotes will suffer the effects, manifesting as embryonic or larval arrest, sterility, or some other phenotype that would have prevented them from contributing to the RILs. The proportion of $F_2$s showing such phenotypes is expected to be around 44% (7/16), assuming that arrest by each toxin is fully penetrant and rescuable by a single copy of the corresponding antidote. Given the presence among RILs of rare NIC58 homozygotes at each of the two transmission ratio distortion loci, penetrance must be less than 100%, and so the proportion of affected $F_2$ progeny should be less than 44%.

To test whether $F_2$ populations from a NIC58 × JU1373 cross showed phenotypes consistent with two Medea-like loci, we made reciprocal crosses, allowed $F_1$ hermaphrodites to self-fertilize, and followed $F_2$ progeny from embryo to adulthood. We observed that far fewer $F_2$ embryos developed into adults within the normal developmental time than expected from the two-Medea model (41–45%, n = 329–1283, versus 98–99%, n = 1046–1093, for parental strains). Terminal phenotypes among abnormal $F_2$ animals included failure to hatch (9%), early larval arrest (39–41%), late larval arrest (5–9%), and abnormally small, thin adults (5%, n = 329–381); these phenotype frequencies did not differ according to the direction of the cross that generated the $F_1$ worm (Fisher's exact test, p=0.39). These data show that $F_2$ populations experienced widespread developmental arrest, at proportions higher than expected under a model of two Medea loci. One interpretation is that the total arrest proportion among $F_2$ individuals reflects the effects of two Medea loci (<44% of $F_2$ animals), plus additional background incompatibilities (>12–15% of $F_2$ animals).

The two-Medea model predicts that developmental arrest among $F_2$ animals will preferentially affect those homozygous for non-Medea (NIC58) alleles. To test this prediction, we repeated reciprocal NIC58 × JU1373 crosses and genotyped $F_2$ progeny (including arrested or dead larvae) at markers tightly linked to the transmission ratio distortion peaks on chromosomes III and V. We observed that alleles at both loci were transmitted to progeny in Mendelian proportions, showing that meiosis and fertilization are not affected by the loci and that distortion is due to postzygotic lethality. Certain genotypes were associated with developmental arrest, but these differed depending on the direction of the cross (i.e., mitochondrial genotype). For the chromosome III locus, NIC58 homozygotes underwent complete developmental arrest in the mito-JU1373 cross but very little developmental arrest, compared to other genotypes, in the mito-NIC58 cross (*Figure 6A*). For the chromosome V locus, NIC58 homozygotes experienced highly penetrant developmental arrest in both crosses (*Figure 6A*). Chromosome V JU1373 homozygotes also experienced developmental arrest in both crosses, but with a lower penetrance, especially in the mito-JU1373 cross (*Figure 6A*). This pattern supports a model of two Medea loci, but implies that the loci interact with mitochondrial genotype: for the locus on chromosome III, the Medea allele (JU1373) is active in its own

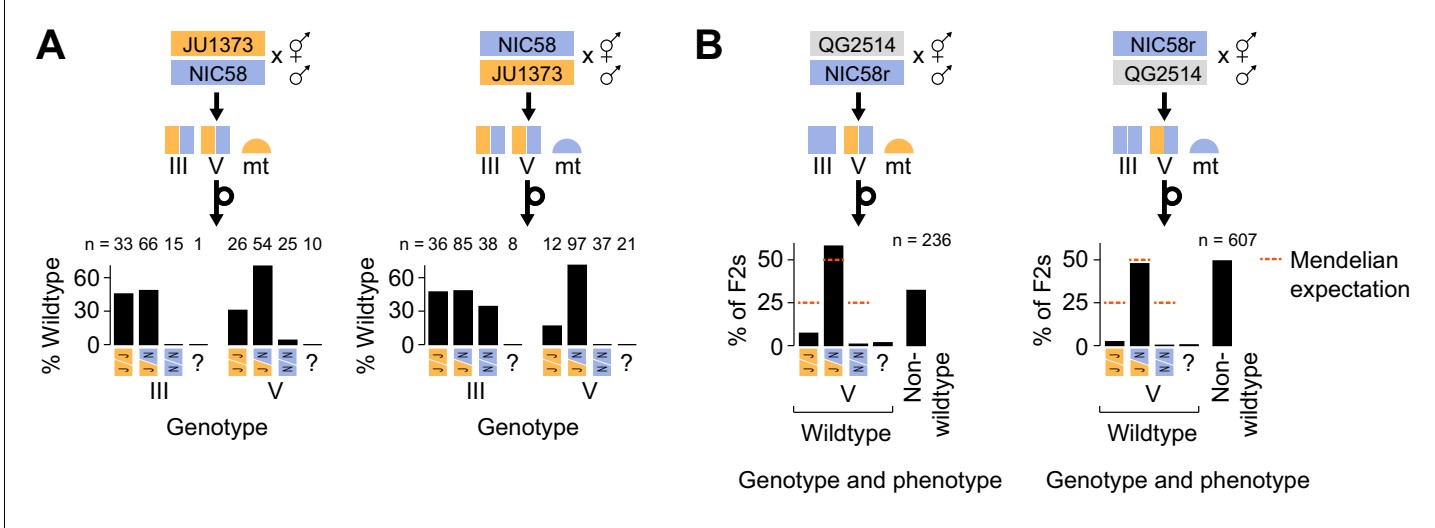

**Figure 6.** Medea genetics. (A) Percentage of $F_2$ progeny from reciprocal NIC58 × JU1373 crosses showing wild-type development. Genotypes reflect markers tightly linked to the peaks of transmission ratio distortion on chromosomes III and V. ? = genotyping failure. (B) Genotype and phenotype frequencies among $F_2$ progeny from reciprocal crosses between NIC58 and recombinant inbred line (RIL) QG2514. Only wild-type $F_2$ progeny were genotyped. Data are in *Figure 6—source data 1*.

The online version of this article includes the following source data for figure 6:

**Source data 1.** NIC58_JU1373_RIL_crosses.tsv.zip; plate-level cross compatibility data for JU1373, NIC58, and RILs.

mitochondrial background but inactive or very weakly active in the opposite mitochondrial background; for the locus on chromosome V, both alleles (JU1373 and NIC58) act as Medea elements and are effectively antagonistic – JU1373 acts strongly in both mitochondrial backgrounds, whereas NIC58 acts strongly in its own mitochondrial background and more weakly in the opposite mitochondrial background (*Figure 6C*). Importantly, this finding of antagonistic haplotypes at the chromosome V locus provides a simple explanation for the retention of heterozygosity at this locus among the RILs, since both classes of homozygous segregants from a hermaphrodite heterozygous for the chromosome V locus arrest (*Figure 5—figure supplement 1*).

## Antagonistic Medea activity does not reflect mitochondrial-nuclear incompatibility

The reciprocal crosses described above showed that developmental arrest of chromosome V JU1373 homozygotes was more penetrant in a mito-NIC58 background than in a mito-JU1373 background. One model for this pattern is that any animal homozygous for the JU1373 chromosome V locus in the mito-NIC58 background suffers a fitness cost due to a mito-nuclear incompatibility, independent of Medea effects. Alternatively, there is no such intrinsic incompatibility, and instead the mitochondrial background modifies the penetrance of the NIC58 chromosome V Medea, with effects manifest in the offspring of chromosome V heterozygotes. The fitness of JU1373 V homozygotes in a mito-NIC58 background should be reduced under the mitochondrial-nuclear incompatibility model, independent of the genotypes of their parents. We therefore attempted to isolate this genotype by crossing males of RIL QG2514 to NIC58, producing an $F_2$ population segregating at the chromosome V locus but fixed for the NIC58 haplotype at the chromosome III locus (*Figure 6B*). We recovered 10 $F_2$ animals homozygous for the JU1373 V allele in a mito-NIC58 background. These animals produced progeny that were superficially wild type, with typical brood sizes and developmental rates. This finding excludes simple mitochondrial-nuclear incompatibility as a contributor to developmental arrest of chromosome V JU1373 homozygotes. We conclude that differences in arrest according to mitochondrial genotype reflect greater activity of the NIC58 Medea on chromosome V in a mito-NIC58 background than in a mito-JU1373 background. We also conclude that activity of the NIC58 Medea in the mito-JU1373 background is likely dependent on additional nuclear background factors, thus accounting for heterozygosity in the RILs being preferentially

retained in certain genetic backgrounds (e.g., in animals with JU1373 alleles on the left arm of chromosome I). Finally, we note that similar conclusions (mitochondrial × parental × zygotic interactions, not simple mitochondrial × zygotic incompatibility) can be inferred for the chromosome III locus, given the wild-type development observed in the subset of RILs homozygous for the NIC58 chromosome III locus in the mito-JU1373 background (*Figure 7A*). Thus, lethality in *C. tropicalis* is not caused by simple mitochondrial × zygotic incompatibilities, although two of the three Medea elements are more active in the mitochondrial background of their parental strain (*Figure 6C*).

## Medea loci act independently

Our model predicts that the Medea loci act independently and are thus genetically separable. To test this prediction, we examined RILs with opposite parental genotypes at the two loci. Each RIL was crossed to an appropriate parental strain to generate $F_2$ animals segregating opposite alleles at one Medea locus but fixed at the other, which we scored for development. To provide a control for (non-Medea) background incompatibilities, we repeated this analysis for $F_2$ populations fixed for JU1373 alleles at both Medea loci. These controls were generated by crossing 10 RILs carrying JU1373 alleles at both Medea loci to a JU1373-derived strain carrying a Dumpy mutation, which allows us to distinguish cross-progeny from self-progeny, a constant issue for JU1373 with its low rate of crossing by hermaphrodites (*Figure 1*). We observed that crosses segregating only the chromosome III Medea in a mito-JU1373 background showed median rates of normal development consistent with Medea activity at a single locus (68–75%, n = 218–240); crosses in a mito-NIC58 background showed no such activity (*Figure 7A*). The arrested progeny in the mito-JU1373 background largely consisted of chromosome III NIC58 homozygotes, as evidenced by these animals being severely depleted among wild-type progeny (26:64:2 JJ:JN:NN genotypes), while the wild-type progeny of the reciprocal cross carried chromosome III genotypes at their expected Mendelian proportions (30:57:31). These data show that the chromosome III Medea is active in the absence of segregation at the chromosome V locus, and confirm that the chromosome III Medea requires a mito-JU1373 genetic background. The lethality associated with chromosome III therefore represents a three-way interaction between parental nuclear genotype, zygotic nuclear genotype, and a mitochondrial locus.

Crosses segregating only at the chromosome V locus showed median rates of normal development that were similar across mitochondrial backgrounds and consistent with antagonistic Medea activity at a single locus (54–61%, n = 203–321, in a mito-JU1373 background; 52–53%, n = 313–345, in a mito-NIC58 background; *Figure 7A*). Arrested progeny largely consisted of NIC58 homozygotes and, to a lesser extent, JU1373 homozygotes, as evidenced by these genotypes being depleted among wild-type progeny (*Figure 7B*). This result shows that antagonistic Medea haplotypes at the chromosome V locus act in the absence of variation at the chromosome III locus. For comparison, crosses segregating no Medeas showed median rates of normal development that were usually higher than 75%, as expected for crosses lacking drive activity (*Figure 7A*); nonetheless, rates of normal development in these crosses were highly variable and sometimes low, with medians ranging from 65 to 100% (n = 163–213), suggesting a contribution from non-Medea background incompatibilities. We conclude that the chromosome III and V Medea loci act independently of one other, and that additional background incompatibilities are widespread and diffuse in the NIC58 × JU1373 cross.

## Medea loci act via maternal effect

The loci we identified in *C. tropicalis* produce transmission ratio distortion via an interaction between parent and offspring genotypes. To distinguish between maternal and paternal effects, we scored rates of normal development among progeny in which Medea alleles segregated from mothers or fathers. We used a combination of sperm depletion and visible markers to prevent self-progeny from contaminating counts of cross-progeny. For a maternal effect (and no paternal effect), 50% of cross-progeny lacking the Medea were expected to undergo developmental arrest when the Medea was present in the mother, assuming full penetrance, but develop normally when the Medea was present in the father. In the case of a paternal effect (and no maternal effect), the opposite pattern was expected. Crosses testing the JU1373 and NIC58 Medeas showed that all acted via maternal,

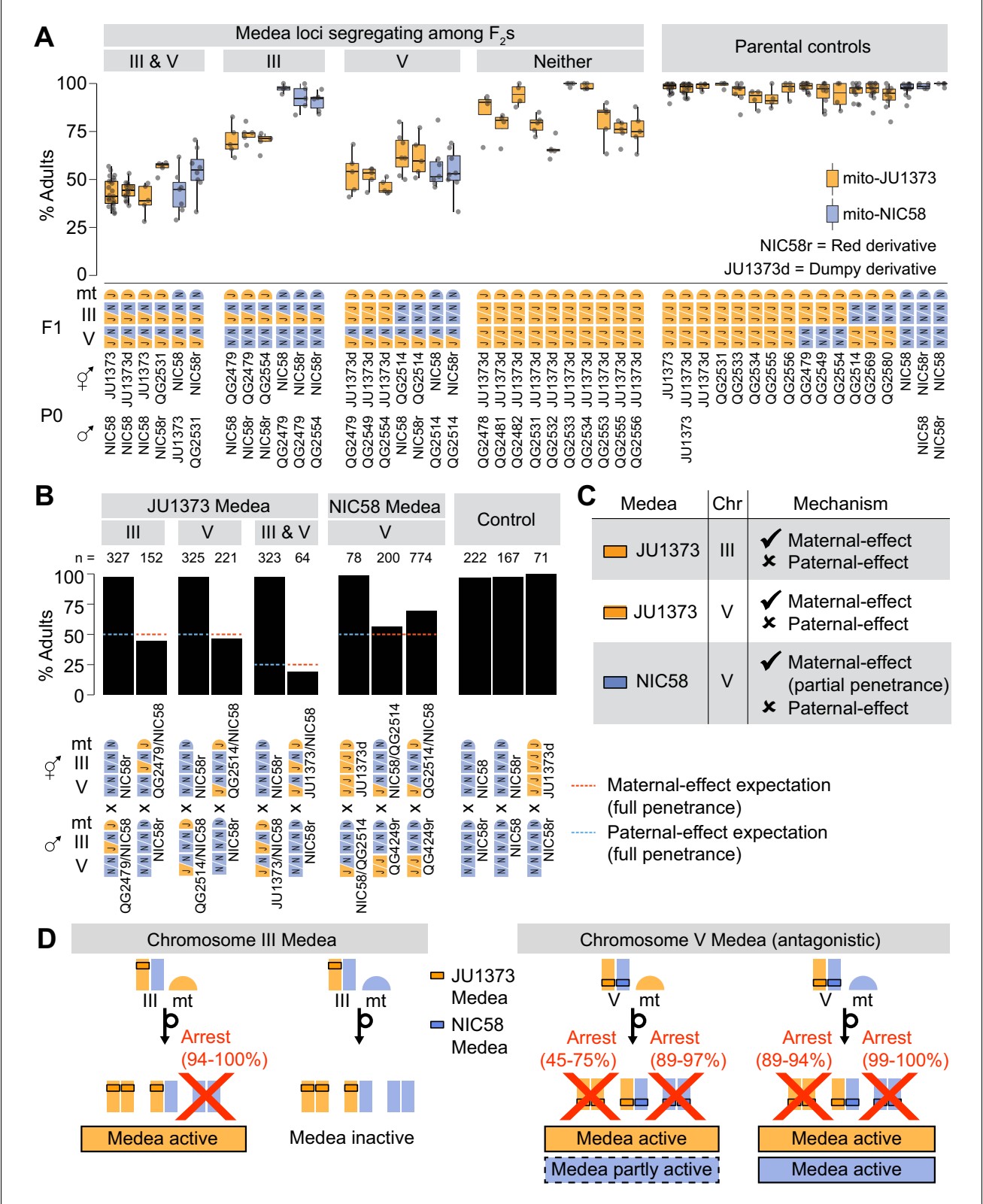

**Figure 7.** Medea loci act independently and by maternal effect. (**A and B**) Percentages of F$_2$ or backcross progeny that reached adulthood within 72 hr of egg laying. Strains beginning with 'QG' are recombinant inbred lines (RILs). JU1373d is a Dumpy mutant. NIC58r and QG4249r express a red fluorescent transgene (see Materials and methods ). (**A**) Crosses testing whether Medea loci act independently. Each point is a cross plate, progeny of a single F$_1$ hermaphrodite, with a median of 34 worms scored per plate. (**B**) Crosses testing whether Medea loci act via maternal or paternal effect.
*Figure 7 continued on next page*

Figure 7 continued

Maternal- and paternal-effect expectations are under a model that either a maternal- or paternal-effect toxin causes fully penetrant developmental arrest for progeny not inheriting the Medea haplotype. (C) Interpretation of maternal- and paternal-effect crosses. (D) Schematic of Medea activity. Percentages are estimates for the proportion of animals undergoing developmental arrest, compared to heterozygous siblings, derived by comparing observed genotype frequencies among wild-type F$_2$ progeny to Mendelian expectations. This method avoids bias introduced by genotyping failures being more common among arrested versus wild-type animals. Reciprocal crosses used to estimate arrest proportions were NIC58 × JU1373, NIC58 × RIL QG2479 (not shown), and NIC58r × RIL QG2514. NIC58r expresses a red fluorescent transgene (see Materials and methods).

The online version of this article includes the following figure supplement(s) for figure 7:

**Figure supplement 1.** Mapping Medea loci using recombinant inbred lines (RILs).

and not paternal, effect (*Figure 7B*). The overall model is presented in *Figure 5—figure supplement 1* and *Figure 7D*.

## Incompatibilities are associated with duplicated and novel genes

To better understand the genetic basis for Medea activity, we mapped the JU1373 elements using RILs with recombination breakpoints near the peaks of transmission ratio distortion. RILs were crossed to derivatives of parental strains, and Medea activity was assessed by scoring the rate of normal development among F$_2$ progeny. This analysis mapped Medeas to intervals of around 33 kb on chromosome III and 69 kb on chromosome V (*Figure 7—figure supplement 1*) that overlapped the peaks of transmission ratio distortion (*Figure 5*). The chromosome III interval encompasses a locus of locally elevated sequence divergence between NIC58 and JU1373, in an otherwise well-conserved region. A clear candidate is an insertion of sequence in JU1373 that includes seven predicted genes (*Figure 8A*): six are tandem duplications of neighboring genes in NIC58; the seventh is a duplication of a gene located 0.68 Mb to the right in both NIC58 and JU1373. Five of the seven genes have no detectable protein or nucleotide homology outside *C. tropicalis*. The remaining two are homologous to *C. elegans* genes *F44E2.8* and *F40F8.11*, which share NADAR and YbiA-like superfamily protein domains. In addition to these seven genes, JU1373 also carries an eighth gene inserted within the original copy of the duplicated sequence (*Figure 8A*, arrowhead). This gene is novel and has no detectable homology to any gene in NIC58 or in any other species. Thus, the JU1373 Medea on chromosome III contains a total of eight additional genes compared to NIC58: one unique to JU1373, and five with no homology outside *C. tropicalis*.

The chromosome V locus lies in a region of high divergence between NIC58 and JU1373, extending well beyond the mapped interval (*Figure 5B*). This region is home to a number of dynamically evolving gene families (*Figure 8B*). A major structural difference between JU1373 and NIC58 haplotypes is an expansion of divergent F-box-domain-encoding genes, expanded from three homologs in NIC58 to 13 in JU1373. Immediately flanking this expansion in JU1373 is a duplication of a gene located 0.6 Mb away (and present in both NIC58 and JU1373), which encodes a homolog of the checkpoint kinase *chk-2*. Adjacent to the *chk-2* homolog is a tandem duplication of a nuclear gene encoding a mitochondrial ubiquinol-cytochrome c oxidoreductase complex protein, perhaps accounting for the interaction of Medea activity with mitochondrial genotype. Additional differences between haplotypes include a gene encoding a small, novel protein in JU1373 and a novel gene in NIC58. Thus, the JU1373 Medea on chromosome V resides within a larger genomic window of elevated sequence divergence, containing an expansion of F-box genes, a novel gene, as well as a duplicated gene encoding a mitochondrial protein. The inferred antagonistic Medea in NIC58 may also involve novel protein-coding potential.

We also examined the mitochondrial genomes of JU1373 and NIC58, and found that the core functional complement is superficially identical; all 12 protein-coding, two ribosomal RNA, and 22 tRNA genes are called as present and intact in both (*Bernt et al., 2013*; *Lemire, 2005*; *Li et al., 2018*). There are, however, several differences of unknown significance, including 12 missense variants in six of the core protein coding genes, the presence in NIC58 of a small non-coding region and a potentially novel (lowly expressed) gene encoding a 2-pass transmembrane protein, and differential expression at two additional loci (*Figure 8—figure supplement 1*). These are additional candidate variants for the observed nuclear-cytoplasmic interaction.

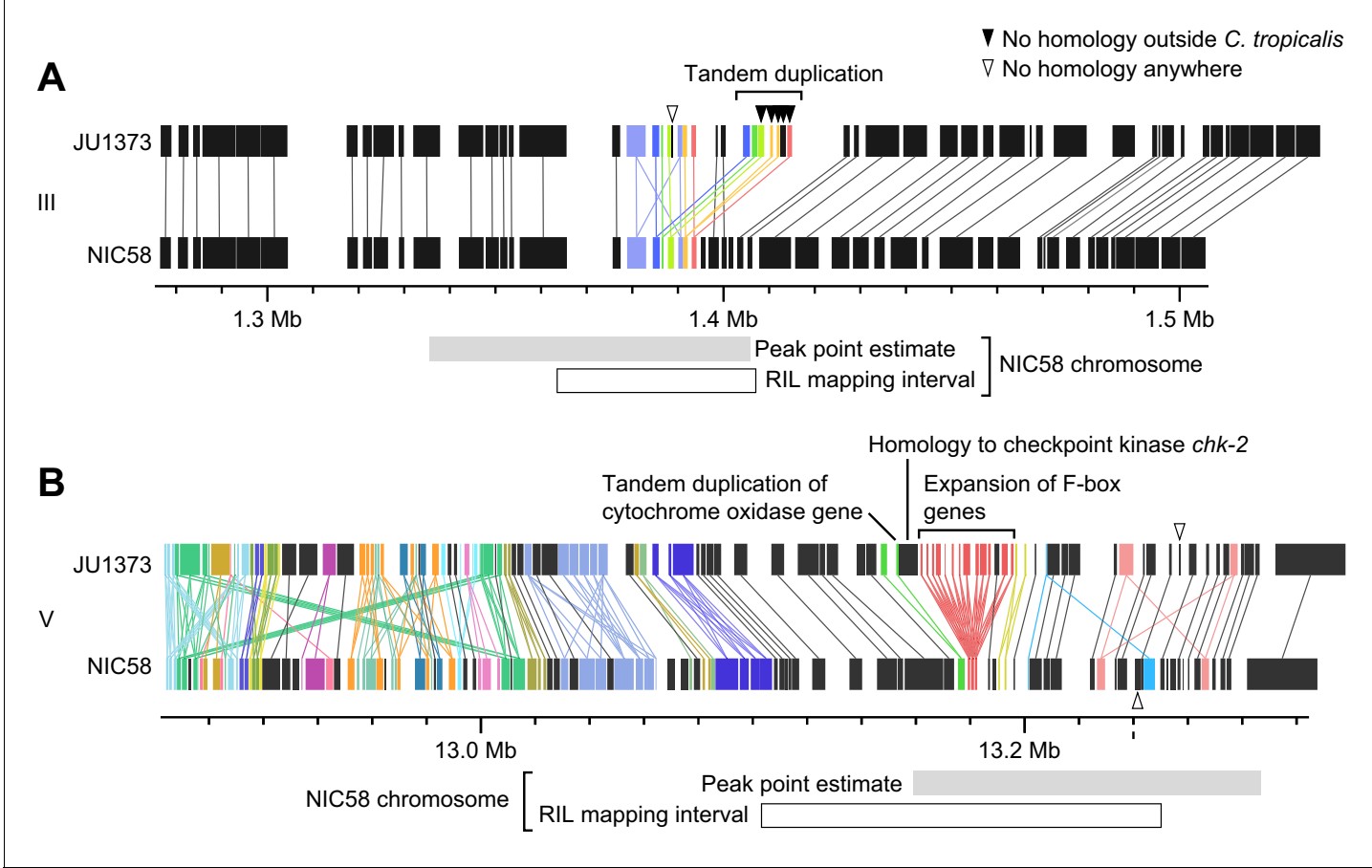

**Figure 8.** Genomic internals surrounding Medea loci on chromosome III (**A**) and V (**B**). Windows span transmission ratio distortion 1 LOD drop intervals. Rectangles are predicted genes, lines connect homologs. Colors indicate the union of homologs within the interval. Homology relationships to genes outside the intervals are not shown. Data are based on *Supplementary file 2* and *Supplementary file 6*.

The online version of this article includes the following figure supplement(s) for figure 8:

**Figure supplement 1.** Mitochondrial genomes.

## Incompatibilities consistent with Medea activity are widespread among wild isolates

To examine the distribution of putative Medea activity among wild strains, we crossed 14 isolates to a Dumpy derivative of JU1373 (JU1373d), to NIC58, or to both, and scored development among the $F_2$. Crosses were classified as having putative Medea activity if rates of normal development were below ~75% (the expectation for segregation of a single Medea). We observed that crosses to JU1373d showed putative Medea activity for 12 of 13 isolates (*Figure 9A*), and the strength of activity was associated with geographic origin, and haplotype at the chromosome III and V Medea loci (which are themselves associated due to population structure). The two African isolates showed the least activity and had haplotypes similar to JU1373 at both Medea loci; five American isolates showed consistently higher activity and carried haplotypes dissimilar to JU1373 at both Medea loci; isolates from other areas were more variable, but activity was lowest for isolates carrying haplotypes similar to JU1373 at the chromosome V locus (NIC535 and NIC773) or somewhat similar to JU1373 at both loci (QG131; *Figure 9B*). Nonetheless, the correlation between Medea activity, geographic origin, and haplotype was imperfect (e.g., JU1630 versus JU3170), and some isolates showed median rates of normal development consistent with segregation of three or more Medeas (NIC517 and QG834, 20–28%, n = 257–276). This pattern suggests that the JU1373 Medeas may be fixed

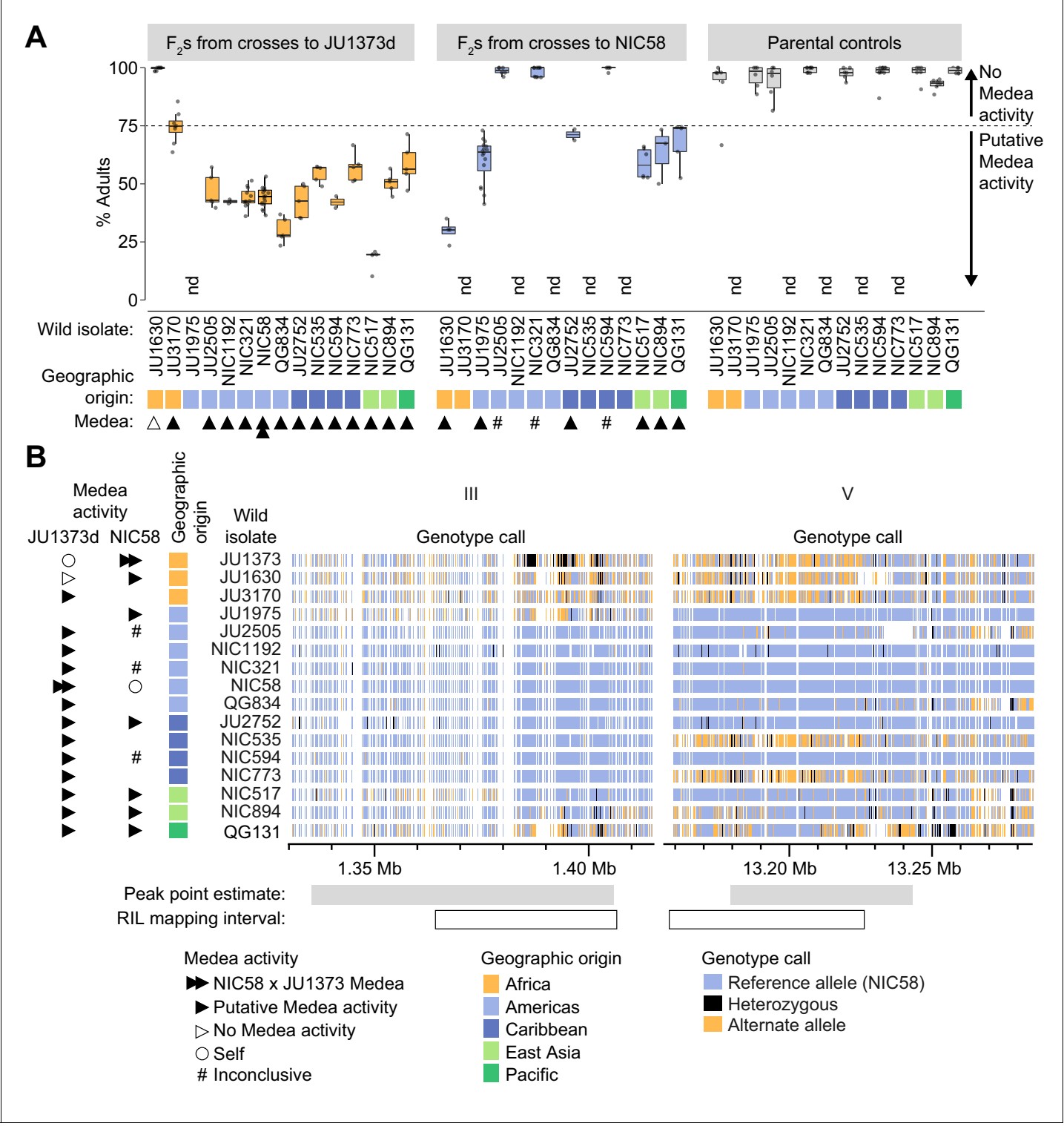

**Figure 9.** Wild isolate phenotypes and haplotypes. (**A**) Percent of $F_2$ progeny reaching adulthood within the normal developmental time (~72 hr), for crosses between wild isolates and a Dumpy derivative of JU1373 (JU1373d) or NIC58. #, inconclusive because we cannot be certain that $F_1$ animals were cross-progeny. nd, not determined. Putative Medea activity is inferred for crosses in which the median percent of $F_2$ animals reaching adulthood was less than ~75%. Each plotted point is a plate (2–16 per cross, median 6), with a median 39 animals scored per plate. Data is in *Figure 9—source data 1*. (**B**) Wild isolate single nucleotide variant haplotypes at the Medea loci on chromosomes III and V. Heterozygous calls likely reflect duplication and divergence. Data are based on *Supplementary file 4*.

The online version of this article includes the following source data for figure 9:

**Source data 1.** NIC58_JU1373_isolate_crosses.tsv.zip; plate-level cross compatibility data for JU1373, NIC58, and wild isolates.

within Africa but polymorphic or absent elsewhere, and that some of the non-African isolates may contribute additional Medeas of their own.

Among crosses to NIC58, putative Medea activity was observed in six of nine crosses (*Figure 9A*). Five of these crosses showed activity that was moderate in strength and largely overlapping, despite these isolates having different haplotype combinations at the chromosome III and chromosome V Medea loci. The sixth cross (JU1630) showed much higher activity, consistent with this isolate alone having JU1373-like haplotypes at both Medea loci. The three remaining crosses showed no drive activity and had haplotypes similar to NIC58 at both loci; we conservatively interpret these crosses as inconclusive because we cannot be certain of paternity for $F_1$ animals (crosses to NIC58 did not include a marker to distinguish cross- from self-progeny). Nonetheless, these data as a whole indicate that putative Medea activity is widespread: Medeas sometimes segregate within a geographic region, and some crosses may segregate additional elements to those identified in the NIC58 × JU1373 cross.

## Medea dynamics in partial selfers

Populations segregating Medea loci like those we discovered on chromosomes III and V are expected to evolve suppressors to alleviate the reduced fitness of animals whose progeny are killed. A general resistance mechanism for suppressing this lethality is selfing, which reduces the prevalence of heterozygotes and hence of animals expressing Medea-associated lethality. To better understand how the mating system of *C. tropicalis* influences the spread of these elements, we simulated evolution under different levels of selfing and outcrossing. These simulations included incomplete penetrance as well as idiosyncratic features of *Caenorhabditis* androdioecy: hermaphrodites cannot mate with one another and males can arise spontaneously by nondisjunction of the X chromosome (see Materials and methods). Selfing rate was a fixed parameter in this system, and we leave a fuller exploration of the coevolutionary dynamics of selfing and intergenerational incompatibilities to future work. We found that in a large population, high rates of selfing can dramatically slow the rate of spread of a Medea element, and indeed can reduce its efficacy such that it fails to increase in frequency during 100 generations of simulated evolution (*Figure 10A*). In small populations, elements introduced at 5% frequency are rapidly fixed under random mating but lost to drift under high rates of selfing (*Figure 10B*). These simulations also reveal that once an element is at high frequency, selfing actually hastens fixation of the Medea element (compare partial selfing at $S = 0.25$ to obligate outcrossing at $S = 0$). This occurs because once an element is at high frequency, heterozygotes that self will always expose their progeny to the element's killing effects, but heterozygotes that outcross will usually mate with an antidote-carrying male, slowing the fixation of the element. These results show that partial selfing slows or prevents the spread of Medea elements at low frequency but hastens their fixation at high frequency.

We next considered antagonistic Medeas, under the simplistic scenario of perfect linkage and equal penetrance. We found that intermediate levels of selfing cause frequency-dependent selection against the rarer haplotype (*Figure 10C*, lower panel, and *Figure 10—figure supplement 2A–C*). Surprisingly, allele frequencies of the antagonistic Medeas drift as though neutral under both obligate selfing and obligate outcrossing, albeit for different reasons. With obligate selfing, there are simply no heterozygotes and the alleles are literally neutral. Under obligate outcrossing, overdominance generated in the offspring generation by the killing of both homozygote classes is offset by underdominance in the parental generation, because the costs of these deaths are borne by heterozygote mothers. The net effect is that the alleles exhibit drift-like dynamics despite the enormous selective cost. Effective neutrality in this scenario depends on the equal penetrance of antagonistic haplotypes in the model. When we model unequal penetrances, similar to those we observe at the chromosome V locus (0.95 and 0.6), we find that a resident strong Medea will prevent the spread of a weaker invader, as expected. When a strong Medea invades a population with a weaker resident Medea, the strong haplotype spreads if outcrossing rates are high but is eliminated under moderate to high rates of selfing (*Figure 10—figure supplement 2D,E*). Overall, selfing reduces Medea load both by decreasing heterozygote frequency and by inducing strong positive frequency-dependent selection that prevents antagonistic alleles from co-occurring.

Finally, we considered the metapopulation biology of *Caenorhabditis* nematodes, which involves colonization of ephemeral habitat patches, rapid population expansion, and then generation of new dispersal morphs (dauers) when the population exhausts its patch (*Cutter, 2015*; *Félix and*

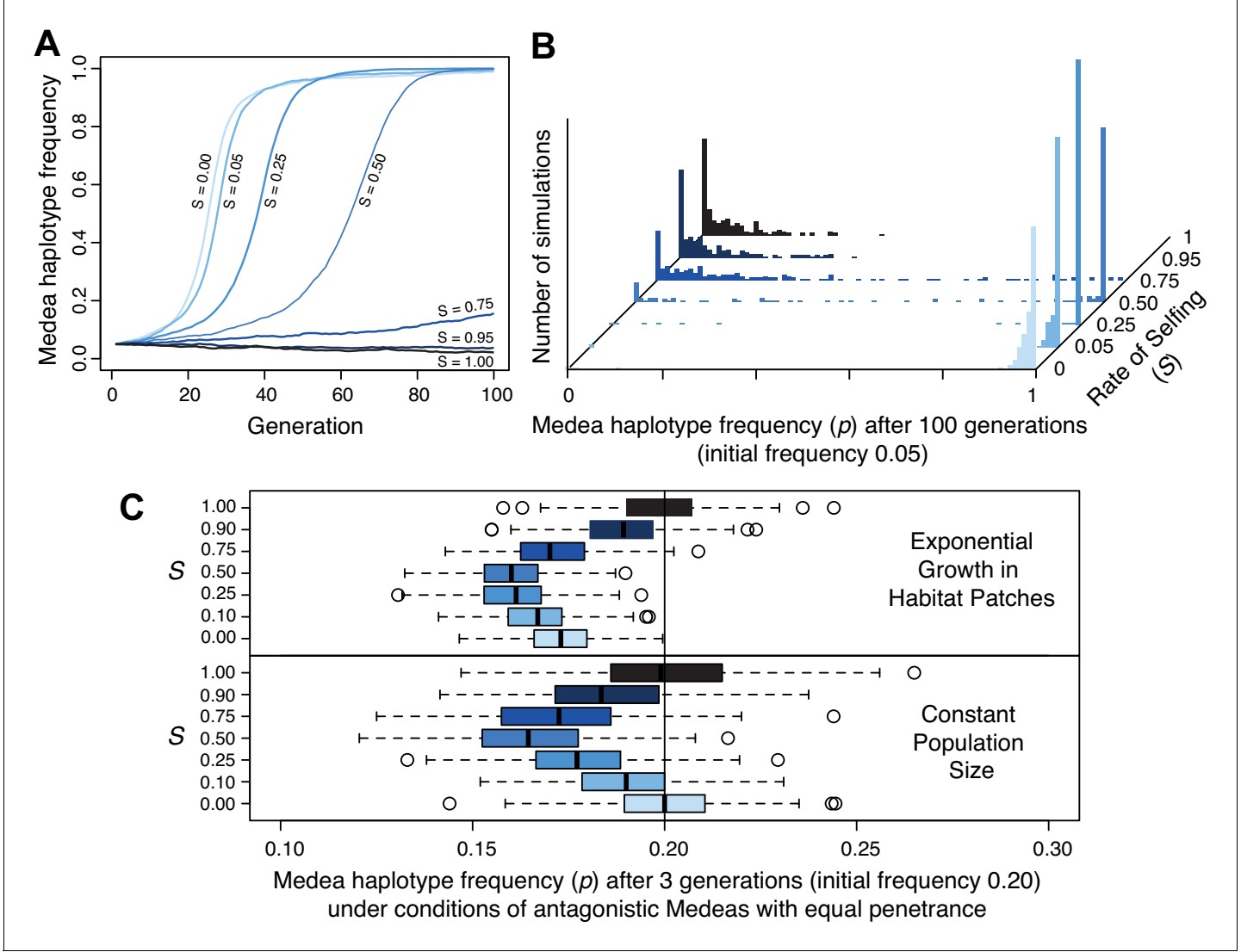

**Figure 10.** Selfing reduces the efficacy of Medea elements. (**A**) Representative allele frequency trajectories of a Medea haplotype under different rates of selfing (S) for 100 generations. Population size is 20,000 in each case, the initial Medea allele frequency is 0.05, penetrance is 0.95, and initial genotype frequencies and sex ratios are those expected at neutral equilibrium given the selfing rate. At selfing rates of 95%, elements fail to increase in frequency during 100 simulated generations. (**B**) Distribution of Medea allele frequencies after 100 generations in populations of size 1000. Each histogram shows the outcome of 250 simulations with initial Medea allele frequency 0.05. Medea alleles are often lost under high selfing rates. (**C**) Antagonistic haplotypes induce positive frequency dependent selection when selfing rates are intermediate, when populations undergo exponential growth in ephemeral habitat patches, or both. Each boxplot represents the results of 250 simulations of three generations of evolution with competing Medea haplotypes starting from allele frequency 0.2 and 0.8, with initial population size 1000. In the patchy environment, those 1000 individuals are distributed among 250 separate patches, and population growth is unbounded within each. Source code is available from github.

The online version of this article includes the following figure supplement(s) for figure 10:

**Figure supplement 1.** In a single-Medea scenario, genic and group selection affect the frequencies of haplotypes, and Medea-induced deaths generate individual-level selection for suppression.

**Figure supplement 2.** Antagonistic Medeas are subject to drift, genic selection, or positive frequency-dependent selection, depending on the selfing rate.

*Braendle, 2010*; *Ferrari et al., 2017*; *Richaud et al., 2018*). If a patch exists for a limited time, patches whose populations expand most rapidly will contribute more descendents to the overall gene pool when the patches expire. Chance genetic differences among patches can strongly influence the rate of population expansion (*Figure 10—figure supplement 1*). Within each patch, allele frequencies experience the same forces as in a population of fixed size, but among patches we

expect that those founded solely by homozygotes (experiencing no selective deaths from Medeas) will have grown the largest after a few generations, increasing the representation of their alleles in the overall gene pool. As the rarer allele at a locus will be overrepresented in heterozygotes, and most homozygotes will carry only the common allele, this mode of population regulation should increase the frequency of the more common allele. This is a kind of 'Haystack Model,' well studied in the context of sex ratios, where patchy environments favor a female bias because of the more rapid population expansion it allows (*Bulmer and Taylor, 1980*; *Wilson and Colwell, 1981*). These group-selection models are sensitive to parameters, including the number of individuals that found a patch, the number of offspring per parent, and the number of generations of growth within each patch. For each selfing rate, we modeled a situation in which each patch is colonized by four individuals, with sex ratio fixed to exclude its effects but genotypes drawn from the equilibrium frequencies given the allele frequency and selfing rate. The population then grows without any constraints on size until the patch reaches its expiration date, after three generations, at which time we calculate the global allele frequency, summing across the entire collection of patches. Under these conditions, we confirm a strong negative relationship between patch heterozygote frequency and patch growth rate. For a single Medea, the effects of growth in a patchy environment on allele frequency are very modest, and the genic selection within each patch allows Medeas to spread despite their costs. With antagonistic Medeas, modeled with equal penetrance, the reduction in heterozygote brood size is greater and the directional effects of single-Medea genic selection are absent. In these conditions, group selection is sufficient to generate a global change in allele frequency (*Figure 10C*). There are more all-homozygote patches for the more common haplotype, and the result is positive frequency-dependent selection, decreasing the frequency of the less common haplotype. Overall, the joint effects of selfing and group selection on antagonistic Medeas are therefore to eliminate the rarer allele. Selection achieves this result most efficiently at intermediate selfing rates. Although the Medeas stick around at higher selfing rates, they impose very little genetic load, and at very high selfing rates they are entirely neutral.

## Discussion

We have brought genetic and genomic resources to the most recently discovered androdioecious *Caenorhabditis* species, *C. tropicalis*, facilitating studies of the varied effects of mating system transitions on genomes and metapopulation genetics, and their interactions with ecology, as well as comparative quantitative genetics. Aided by a chromosome-scale reference genome, our data confirm that *C. tropicalis* is indeed the most genetically homogeneous of the three selfers, on average, which implies a correspondingly high rate of selfing, low rate of effective recombination, and small effective population size. We also show that the average obscures extreme variance in the distribution of genomic diversity. This finding mirrors recent findings for *C. elegans* and *C. briggsae* (*Lee et al., 2020*). Within two highly divergent regions, we find some striking biology: three Medea gene-drive systems segregating in a single cross. Genetically similar Medea elements have also been found in *C. elegans* (*Ben-David et al., 2017*; *Seidel et al., 2008*), but we and others have now shown that they are especially common in *C. tropicalis* (*Ben-David et al., 2020*). They thus represent a potent and surprisingly widespread form of genetic incompatibility underlying outbreeding depression, and a potential cause of the species' high effective selfing rate. Comparative analysis will be furthered by more extensive sampling, in time, and in geographic and genomic space, for all selfing species. This project will also benefit from inclusion of the closely related sister-species of *C. tropicalis*, *C. wallacei* (*Félix et al., 2014*), as in the case of *C. briggsae* and its outcrossing sister *C. nigoni* (*Yin et al., 2018*).

### Selfing and population genetics

Our early view of selfing *Caenorhabditis* species was of widespread, weedy lineages depauperate of genetic diversity relative to their outcrossing ancestors. This view, limited by mostly opportunistic sampling of isolates, and sequencing technology, was based on the apparent global expansion of *C. elegans* associated with human activity (*Andersen et al., 2012*). Better sampling has led to a more complete picture of strong population and genomic structure for all three species (*Crombie et al., 2019*; *Thomas et al., 2015*), though tropical areas remain particularly undersampled.

Most recently, a large survey of *C. elegans* genomes, including 15 assembled from long-reads, found regions of high-diversity spanning up to 20% of the reference genome (*Lee et al., 2020*). Similar heterogeneity was found in a smaller sample of *C. briggsae* genomes. This finding was presaged by efforts to build a complete genome for the divergent Hawaiian isolate CB4856 using second generation sequencing (*Thompson et al., 2015*), but has been greatly enabled by more contiguous assemblies that circumvent the reference mapping bias plaguing study of all genetically diverse species. A promising hypothesis for the presence of hyperdivergent regions in genomes is the action of balancing selection across the species' range, leading to preservation of some of the presumably abundant genetic diversity of outcrossing ancestral species. This hypothesis is supported in *C. elegans* by the enrichment of genes encoding environmentally responsive sensory factors, which are themselves enriched for differential expression and quantitative trait loci for response to microbes isolated from natural habitats. Alternative hypotheses include introgression, a common source of islands of divergence in other animal taxa (*Hedrick, 2013*), but the presence of multiple distinct divergent haplotypes in *C. elegans* (*Lee et al., 2020*) argues strongly against it. Balancing selection can act at many levels, from local adaptation via directional selection, at the global scale, to various frequency-dependent phenomena at the local scale, and overdominance at the molecular scale. While some environmental associations clearly play a role in the biogeography of all three androdioecious species (*Crombie et al., 2019*; *Cutter et al., 2006*; *Dolgin et al., 2008*; *Félix and Duveau, 2012*; *Ferrari et al., 2017*; *Kiontke et al., 2011*; *Prasad et al., 2011*; *Thomas et al., 2015*), they seem not to be a definitive factor in structuring divergent haplotypes. The high migration rate that comes with the microscopic nematode form, coupled with the reproductive assurance afforded by selfing, may effectively counter adaptation to global environmental variation.

Taxa with mixed selfing and outcrossing, very large population sizes, and broad, diverse ranges, may occupy a population genetic space particularly well suited to the detection and localization of balancing and local selection. The ability to detect targets of strong balancing selection scales with total population size (assuming symmetric migration) and recombination rate, and the homogenizing effect of partial selfing is to increase the signal of balanced peaks against a background continually swept of diversity by indirect selection (*Charlesworth et al., 1997*; *Nordborg et al., 1996*). The particularly high selfing rate of *C. tropicalis* may be especially favorable, and its preferred habitat may also allow for more rapid evolution than the temperate-dwelling *C. elegans*. Detection of the targets of local adaptation depends on population size, migration rate, and selection intensity (*Charlesworth et al., 1997*), which are all unknowns. Although Medea factors may have the potential to generate balancing selection, it is clear that most divergent regions between NIC58 and JU1373 do not carry Medea factors, or, at least, not ones that are active independent of environment and genetic background. These, and other highly divergent haplotypes in *C. tropicalis*, may harbor loci under positive selection in different conditions, a conjecture made more plausible by the analysis of divergent gene content in *C. elegans*.

## Population dynamics of Medea elements

Medea elements like those we discovered in *C. tropicalis* will reliably spread in randomly mating populations (*Wade and Beeman, 1994*). Highly penetrant killing of homozygous larvae is an exceptionally potent selective mechanism to drive allele frequency change, and in our short RIL construction pedigree we saw two haplotypes sweep nearly to fixation. In nature, things are likely quite different, as both the mating system and natural history of *C. tropicalis* conspire to render Medea loci selectively inert. *C. tropicalis* shows strong geographic structure, presumably exacerbated by habitat fragmentation, such that encounters between Medea and sensitive haplotypes may be rare. Upon an encounter between a hermaphrodite and a male, outcrossing rates, measured under benign laboratory conditions, are relatively low on average. When divergent isolates do cross, Medeas may find themselves inactive due to dependence on genetic background, including mitochondrial genotype, and, potentially, environmental factors. Most importantly, *C. tropicalis* reproduces primarily by self-fertilization, and gene-drive elements are unable to gain traction when heterozygotes are infrequent. At the same time, the patchy, ephemeral microhabitat of *C. tropicalis* – rotting fruits and flowers on the forest floor – provides a perfect substrate for group selection. Small numbers of dispersing larvae colonize each patch and undergo exponential population growth for a small number of generations. Although Medeas will increase in frequency in patches with

heterozygotes, population growth in patches without heterozygotes can be so much greater as to overwhelm the countervailing effects of Medeas on allele frequency.

The patterns we observe on chromosome V implicate tightly linked Medeas, one on each of the alternative haplotypes, a phenomenon also discovered by *Ben-David et al., 2020*. Surprisingly, we found by simulation that antagonistic Medeas do not generate balancing selection, at least under the scenarios modeled. They nevertheless impose a strong segregation load under outcrossing, which should select for suppressors. The antagonistic Medeas themselves evolve by drift at high selfing rates, and at intermediate selfing rates frequency-dependent selection eliminates the rarer haplotype. At the same time, the chromosome V Medeas occur on ancient haplotypes, evidenced by extreme divergence between NIC58 and JU1373 (*Figure 5*; *Figure 8*). These haplotypes may encode unique toxin-antidote pairs that arose independently, or they may encode toxin-antidote pairs that co-evolved from a common ancestor but are no longer cross-compatible. Competition among driver haplotypes is known to occur for Segregation Distorter in *D. melanogaster*, but in that case, driver haplotypes compete for slots in a balanced equilibrium with non-drivers (*Brand et al., 2015*; *Presgraves et al., 2009*). Our simulations raise questions about whether antagonistic Medeas play a role in the ancient balancing selection at the chromosome V locus, and we note that the two haplotypes are sufficiently different in gene content that effects on other phenotypes are likely.

## Molecular mechanisms of Medea-mediated gene drive

The Medea factors we have discovered in *C. tropicalis* are analogous to the *sup-35/pha-1* maternal-effect driver (*Ben-David et al., 2017*) and the *zeel-1/peel-1* paternal-effect driver in *C. elegans* (*Seidel et al., 2008*; *Seidel et al., 2011*), and four maternal-effect *Medea* drivers in Tribolium (*Beeman et al., 1992*; *Beeman and Friesen, 1999*). Additionally, several other *C. tropicalis* Medea factors have been independently identified and characterized by *Ben-David et al., 2020*. Similar inheritance patterns have also been reported for two loci in mice (*Peters and Barker, 1993*; *Weichenhan et al., 1996*, *Weichenhan et al., 1998*; *Winking et al., 1991*). The causal genes underlying the JU1373 and NIC58 Medeas remain to be identified, but likely include one or more of the multiple genes unique to these haplotypes, as seen for *C. elegans* where toxin and antidote functions are encoded by genes present on the killing haplotype and absent (or pseudogenized) on the non-killing haplotype (*Ben-David et al., 2017*; *Seidel et al., 2008*). Similarly, the single *Tribolium* Medea element whose genetic basis is known maps to a transposable element insertion absent from non-Medea haplotypes (*Lorenzen et al., 2008*). A common pattern emerging from these systems is that maternal-effect drivers (and the single example of a paternal-effect driver) are encoded by dispensable genes with dedicated functions, rather than genes acquiring toxin or antidote activity while retaining an ancestral non-drive function.

Why are Medeas so prevalent in *C. tropicalis* (and to some extent in *C. elegans* and *C. briggsae*) but mostly absent elsewhere? One option is ascertainment bias – maybe similar elements are taxonomically more widespread, but we simply have not looked for them.

A second option is that mechanisms of translational control in the *Caenorhabditis* germline may make it easy for Medeas to arise. Early embryogenesis in *Caenorhabditis* is largely controlled by maternal regulators, and a common expression pattern for these regulators is ubiquitous expression of mRNA in the oocyte but no translation until embryogenesis (*Evans, 2005*; *Robertson and Lin, 2015*). A gene whose protein is generally cytotoxic could become a maternal-effect toxin by acquiring the (common) regulatory elements specifying this expression module. If such a mutation occurs in tight linkage to an incidental zygotically expressed antidote, it creates a Medea. Variation among species in maternal gene regulation may therefore be relevant to variation in the mutational flux of novel Medeas.

The antidote-first scenario, in which Medeas arise from sequential mutations on a single haplotype, may be facilitated by high rates of selfing. In populations with high selfing rates, alternate haplotypes rarely encounter one another in heterozygotes. Each can then evolve as though in allopatry with the other. If balancing selection preserves the alternate haplotypes for a long time, they may incidentally accumulate antidotes and toxins that are neutral in their own backgrounds (*Seidel et al., 2008*).

Toxin-first evolution is also possible, aided by metapopulation structure and selfing. Partially penetrant toxins might arise in a background lacking an antidote but become locally fixed, despite their deleteriousness, due to the tiny effective population size and inefficient purifying selection of a local

selfing population. If outcrossing is rare, toxin-free haplotypes will not be reintroduced or decoupled by recombination and instead, the population might restore its fitness via compensatory evolution of an antidote; if the antidote is linked to the toxin, a Medea is born.

Whether the toxin-antidote elements in *C. tropicalis* and *C. elegans* arose before the transition to selfing is unclear, although the level of divergence between opposite haplotypes at Medea loci is suggestive of sampling from outcrossing ancestors. A closer examination of gonochoristic species in the Elegans group is needed to determine whether toxin-antidote elements are specific to, or quantitatively different in, selfers. It will also be interesting to see the strong outbreeding depression in other taxa with mixed-mating, such as the 'cryptic biological species' complexes in arctic Draba (*Grundt et al., 2006*), dissected genetically.

## Strategies to combat Medea factors

Gene drive systems create a selective environment favoring the evolution of suppressors. Suppressors of meiotic and gametic drive have been well documented in many species, especially when obligate outcrossing continually exposes individuals to the costs of drive (*Courret et al., 2019*; *Lindholm et al., 2016*; *Lyttle, 1991*; *Price et al., 2019*). Suppressors of drive in selfing species are more rare, which has been interpreted as evidence that many drivers in selfing species did not evolve as drivers per se but instead evolved through non-drive mechanisms (*Sweigart et al., 2019*), such as balancing selection maintaining alternate homozygous genotypes (*Seidel et al., 2008*). Our data show that unlinked modifiers affect Medea activity in *C. tropicalis*, though whether these modifiers evolved as suppressors is equivocal. In the case of maternal-effect drivers, mitochondrial suppressors are special: selection for mitochondrial suppressors may be especially strong because mitochondria cannot segregate away from drivers via inheritance in sperm. This selective environment may explain why two of the Medeas we discovered in *C. tropicalis* (the chromosome III JU1373 Medea and the chromosome V NIC58 Medea) were differentially active according to mitochondrial genotype – the mitochondrial genotypes non-permissive for Medea drive may have evolved as suppressors. Alternatively, Medeas may have arisen in permissive mitochondrial backgrounds, with little selection for or against alternate mitochondrial genotypes. Ultimately, our data provide little conclusive evidence that the Medea loci experience selection in nature that is due to their drive activity.

The data suggest that crosses between geographically distant *C. tropicalis* isolates will typically reveal multiple Medea loci (*Figure 9*). Segregation of multiple Medeas magnifies the cost of outcrossing and reduces the possibility of suppression by a common molecular mechanism, in a manner analogous to the role of multidrug therapy in preventing the evolution of drug resistance. The difficulty that organisms face in evolving suppressors to multiple drive elements at once has emerged as an important consideration for gene drive strategies for controlling disease vectors (*Burt, 2003*; *Champer et al., 2018*). In such cases, organisms can adapt by altering their population biology, increasing their rates of inbreeding and selfing (*Bull, 2016*; *Bull et al., 2019*; *Drury et al., 2017*), and thus reducing the heterozygosity required for all gene drive activity.

The costs of selfing as a defense against gene drive are inbreeding depression; reduced ability to adapt to new conditions; and reduced genetic variation and hence niche breadth. Androdioecious *Caenorhabditis* appear to have mechanisms for dealing with each of these costs. Selfing *Caenorhabditis* are typically found in nature as totally inbred lines, consistent with having purged recessive deleterious variants in their history (*Anderson et al., 2010*; *Richaud et al., 2018*). While outcrossing plays an important adaptive role in selfing *Caenorhabditis* (*Chelo et al., 2019*; *Morran et al., 2009*; *Teotónio et al., 2006*; *Teotonio et al., 2012*), populations can transiently increase male frequency to achieve adaptation before returning to a primarily selfing mode of reproduction (*Anderson et al., 2010*; *Shi et al., 2017*). Finally, the preservation of genetic diversity at large numbers of ancient haplotypes by balancing selection allows these species to occupy a wide range of habitats despite low levels of baseline genetic variation (*Lee et al., 2020*).

We have shown that *C. tropicalis* harbors abundant heritable variation in outcrossing rate, with nondisjunction, male mating propensity, and hermaphrodite mating propensity all providing avenues for genetic fine-tuning of the outcrossing rate. Other data also show that *C. tropicalis* has mechanisms that promote selfing over outcrossing. *Ting et al., 2014* found that *C. tropicalis* hermaphrodites are uniquely resistant to the deleterious effects of interspecific matings, and they interpret their findings as evidence for reduced activity of sperm guidance cues in *C. tropicalis* hermaphrodites. *Shi et al., 2017* showed that male longevity is reduced in *C. tropicalis* when male pheromone

is present, creating a negative feedback that tamps down male frequencies, but not in obligately outcrossing *Caenorhabditis* species. These findings are consistent with selection favoring high selfing rates in *C. tropicalis*.

Selfing is often considered a factor that favors the evolution of incompatibilities and outbreeding depression, just as the independent evolution of species or subspecies often leads to incompatibilities revealed by hybridization (*Fishman and Sweigart, 2018*; *Maheshwari and Barbash, 2011*; *Presgraves, 2010*). Selfing reduces the effective recombination rate, allowing unlinked loci to evolve together. When outcrossing reshuffles these co-evolved loci, it creates new combinations of alleles untested by selection. Incompatibilities between these alleles manifest as outbreeding depression (equivalently, rearrangements can fix within selfing lineages, rendering outbred progeny deficient). Our findings suggest causation running in the opposite direction should also be considered. Incompatibilities in *C. tropicalis* appear to mostly represent interactions between tightly linked loci acting in different individuals (mothers and offspring), rather than interactions between unlinked loci; thus, the resulting outbreeding depression is mostly not mediated by recombination. *C. tropicalis* can escape these incompatibilities and restore fitness by inbreeding. Thus, in contrast to the usual pattern of selfing leading to incompatibility, in this species incompatibility may also lead to increased selfing.

## Materials and methods

### Strain maintenance
Strains were maintained using standard protocols for *C. elegans* (*Brenner, 1974*; *Stiernagle, 2006*), with the addition of 1.25% agarose to NGM-agar (NGMA) plates to discourage burrowing, and a 25° C incubation temperature. This temperature is characteristic of substrate temperatures where we have collected *C. tropicalis* and is the standard rearing temperature in previous work on this species (*Gimond et al., 2013*).

### Genome sequencing
Long-read data for NIC58 and JU1373 were around 250× expected coverage, given an estimated genome size of roughly 80 Mb (*Fierst et al., 2015*), from a PacBio Sequel at the Duke University Center for Genomic and Computational Biology. DNA was extracted from twelve 10 cm NGMA plates of nematodes spotted with OP50 using the Qiagen MagAttract HMW DNA kit as per *Lee et al., 2020*.

JU1373 and NIC58 short-read data were around 25× and 40× expected coverage 100 bp paired-end reads (TruSeq libraries, HiSeq 2000, NYU Center for Genomics and Systems Biology Genomics Core), and another 40× coverage for NIC58 (150 bp paired-end reads, TruSeq library, NovaSeq6000, Novogene).

We sequenced 129 RILs from the cross between JU1373 and NIC58 to a median depth of 2.1× (NextEra libraries using the protocol of *Baym et al., 2015*, NextSeq 500, paired-end 75 and 150 bp reads, NYU Center for Genomics and Systems Biology Genomics Core). DNA was isolated by proteinase-K digestion followed by phenol/chloroform/isoamyl alcohol purification.

An additional 22 wild isolates were sequenced to a median depth of 29× (NextEra libraries, NextSeq 500, paired-end 75 bp reads; or BioO libraries, HiSeq 2000, paired-end 100 bp reads; NYU Center for Genomics and Systems Biology Genomics Core). DNA was extracted by salting out (*Sunnucks and Hales, 1996*). Isolates and associated metadata are in *Figure 3—source data 1*.

All sequencing reads used in this project are available from the NCBI Sequence Read Archive under accession PRJNA662844.

### Genome assembly
Our reference genome is NIC58. We generated initial assemblies for evaluation with genetic linkage data, including a Canu hybrid assembly (*Koren et al., 2017*) and long-read only assemblies from flye (*Kolmogorov et al., 2019*), ra (*Vaser and Šikić, 2019*) and wtdbg2 (*Ruan and Li, 2019*). All were initially run with default parameters. Flye produced a highly contiguous assembly with this data, and initial genetic evaluation showed few errors (interchromosomal chimeras were detected for all other assemblers), so we varied parameters (minimum overlap length 4–10 kb, initial assembly depth 40–

180×) and selected the two most contiguous assemblies for closer evaluation (the genetically concordant assembly used -m 10 kb –asm-coverage 120×). A draft assembly for JU1373 was made with flye using default parameters (44 contigs and scaffolds, NG50 4.2 Mb, 81 Mb span). Both assemblies were polished with short-reads using Pilon (-fix bases mode) before further use (*Walker et al., 2014*).

Mitochondrial genomes were initially assembled from long reads mapping to contigs identified as partially homologous to *C. elegans* sequence. De novo assemblies using Unicycler (*Wick et al., 2017*) to produce a polished circular sequence showed homology to all 12 *C. elegans* proteins for both NIC58 and JU1373, but total length and sequenced identity were sensitive to input read length (using all data, or only reads of length 10–15 kb, which spans the range of full-length *Caenorhabditis* mitochondrial genomes in GenBank) and mapping quality. We instead used fragmented, high-coverage contigs from Illumina de novo assemblies (Platanus 1.2.4; *Kajitani et al., 2014*) with homology to the long-read assemblies as bait to extract short reads for reassembly, which produced single sequences of length 13,565 and 13,091 bp for NIC58 and JU1373, respectively. After circular polishing with long-reads (Unicycler), sequences were 14,394 and 14,027 bp. We rotated these with five copies of the *C. elegans* mitochondrial genomes to optimize linear homology using MARS (*Ayad and Pissis, 2017*).

## Genetic map construction

RILs were derived by crossing a NIC58 male and JU1373 hermaphrodite, and inbreeding the $F_2$ offspring of a single $F_1$ hermaphrodite for 10 generations by selfing. We used the RIL data to evaluate assemblies based on interchromosomal consistency and concordance between genetic and physical marker order. A SnakeMake pipeline (*Köster and Rahmann, 2012*) implementing this procedure is on github.

Using short-read mapping to the NIC58 assemblies, we called variants distinguishing the parental lines, filtered them to homozygous diallelic SNVs (depth within 1/3 of the median, >10 bp from an insertion/deletion, quality >50, then removing any SNVs in 20 bp windows with more than one SNV), and genotyped RILs at the remaining sites (*Li, 2011*; *1000 Genome Project Data Processing Subgroup et al., 2009*; *Li and Durbin, 2009*; *Vasimuddin and Misra, 2019*).

Parental ancestry was inferred by HMM (*Andolfatto et al., 2011*), sampling one variant per read, with transition probabilities defined by homozygous priors, recombination rate ($r$ = per base pair rate given an expected six recombination events per RIL genome), physical distance between markers in the reference genome ($d$) and a scaling factor ($rfac = 10^{-11}$), parameterized as $10^{r*d*rfac}$, and emission probabilities set by parental genotyping error rate ($10^{-4}$) and read base quality scores. Markers for map construction were defined by filtering on posterior probability >0.5, binning up to 50 SNVs, and merging the sparse RIL marker inferences, interpolating missing positions across consistent homozygous flanking bins. Bins with both parental genotypes were considered as missing data.

Marker filtering and map construction were carried out in R/qtl (*Broman et al., 2003*). After dropping identically informative genotypes, two lines that were outliers for heterozygosity, and one of each of eight pairs of lines with >99% similarity, linkage groups (LGs) were formed (maximizing the number of markers in six LGs), and markers were ordered within LGs by likelihood from 100 iterations of greedy marker ordering. Where genetic and physical ordering conflicted, the physical order was tested by likelihood and accepted if the change in LOD was >−1. Taking the genetic data as ground truth, we compared assemblies on the number of sequences spanning more than one LG, and on the number and sum of negative LOD scores for any remaining discordance in within-LG genetic/physical marker order.

## Genome orientation and scaffolding

On the above metrics, we selected a flye assembly, spanning 81.83 Mb in 36 contigs and scaffolds >20 kb with an N50 of 4.795 Mb (half the expected genome size of 80 Mb was in sequences of at least this length). The genetic map based on this assembly incorporated 33 of these sequences and spanned 81.3 Mb. We then did two rounds of manual stitching, considering only junctions with estimated genetic gaps of 0 cM. First, we accepted 10 joins where sequences from another assembly spanned a junction (>5 kb of MQ = 60 alignment on either flank; minimap2 *Li, 2018*). Second, we

accepted eight joins where at least one read consistent with the genetic orientation spanned a junction (alignment >2 kb of MQ >20 on each 50 kb flank; minimap2). We took the consensus sequence (bcftools), or in two cases the read sequence, and converted the now fully oriented assembly of 15 sequences into pseudochromosomes, with 50 bp N gaps at the remaining junctions. Chromosomes were named and oriented based on *C. elegans* homology, by summing aligned lengths per chromosome and strand (minimap2 -x asm20, mapping quality >30). Chromosome preference was unequivocal (>60-fold bias toward a single homolog in all cases). Strand preference was relatively strong for chromosomes II and X (>3.2-fold bias), but less so for the others (1.8-fold bias for IV, 1.4 for I, 1.3 for III, and 1.1 for V), from 0.400 to 1.9 Mb of aligned sequence per homologous chromosome. The inferred orientations were consistent with strand bias from 1:1 orthologs in chromosome centers in all cases except chromosome I. Finally, we did one further round of short-read polishing (pilon -fix bases mode, making 5044 changes), and re-estimated the genetic map.

## Annotation

### Mixed-stage RNA preparation

We collected three samples each for NIC58 and JU1373: well-fed mixed-staged (L1-adults), well-fed male-enriched, and starved (including dauers) plates. Strains were passaged by chunking every 2 days to maintain a well-fed mixed-stage population. Some plates were allowed to starve, and the presence of dauer larvae along with other developmentally arrested larvae was confirmed by visual inspection. Crosses were set up on single-drop OP50-seeded plates with 15–20 males and a few hermaphrodites to establish a male-enriched population. Following successful mating, worms were chunked to 10 cm OP50-seeded plates for sample collection.

Each sample was collected from one 10 cm plate, flash frozen in 100 μl S-Basal in liquid nitrogen, and stored at −80°C until extraction. Total RNA was extracted using Trizol reagent (Invitrogen) following the manufacturer's protocol, except that 100 μl acid-washed sand (Sigma) was added during the initial homogenization step. RNA was eluted in nuclease-free water, purity was assessed by Nanodrop (ThermoFisher), concentration was determined by Qubit (ThermoFisher), and integrity was assessed by Bioanalyzer (Agilent). Following quality control, 1.5 μg of total RNA from each sample was pooled, further purified using the RNA MinElute Cleanup kit (Qiagen), and again subject to the above quality control analyses.

### Library preparation and sequencing

RNAseq libraries for JU1373 and NIC58 were prepared simultaneously from mRNA isolated from 1 μg of pooled total RNA using the NEBNext Poly(A) mRNA Magnetic Isolation Module (New England Biolabs). RNA fragmentation, first and second strand cDNA synthesis, and end-repair processing were performed with the NEBNext Ultra II RNA Library Prep with Sample Purification Beads (New England Biolabs). Adapters and unique dual indexes in the NEBNext Multiplex Oligos for Illumina (New England Biolabs) were ligated, and the concentration of each library was determined using Qubit dsDNA BR Assay Kit (Invitrogen). Libraries were pooled and qualified by Bioanalyzer 2100 (Agilent; Novogene, CA, USA), and 150 bp paired-end reads were sequenced on a single Illumina NovaSeq 6000 lane.

### Gene prediction

We identified repetitive sequences in the NIC58 and JU1373 genomes de novo using RepeatModeler (*Smit et al., 2020*) and classified these using the RepeatClassifier tool from RepeatModeler and the Dfam database (*Hubley et al., 2016*). We removed unclassified repeats and soft-masked the genome assemblies with RepeatMasker using the classified repeat library. We aligned short RNAseq reads to the soft-masked genomes with STAR in two-pass mode (*Dobin et al., 2013*), and used the BRAKER pipeline to annotate genes (*Hoff et al., 2019*). We extracted protein sequences from the BRAKER annotation using the getAnnoFasta.pl script from AUGUSTUS (*Stanke et al., 2006*), and assessed biological completeness using BUSCO (*Seppey et al., 2019*). We annotated mitochondrial genomes by homology using the MITOS2 server (*Bernt et al., 2013*) and Prokka (*Seemann, 2014*).

## Population genetics

### Variant calling

A SnakeMake pipeline implementing variant calling and filtering is available from github (*Köster and Rahmann, 2012*). In brief, we mapped reads to the NIC58 reference genome with bwa mem2 (*Vasimuddin and Misra, 2019*), aligned and normalized indels with bcftools (*Li, 2011*), called variants jointly with GATK (*DePristo et al., 2011*; *McKenna et al., 2010*), and hard filtered diallelic SNVs (median absolute deviation in total depth <99$^{th}$ percentile, QD > 4, MQ > 30, BaseQRankSum > −3, ReadPosRankSum > −4, SOR < 5). We also applied per-sample depth filtering (local depth in 1 kb windows < 2× against a LOESS polynomial fit for each chromosome, span = 0.33), keeping SNVs in windows where at least 22/24 samples passed. A total of 880,599 diallelic SNVs were called, 794,676 passed filtering (genotype set 1), and we used the fully homozygous subset of these with no missing data, comprising 397,515 SNVs (genotype set 2).

### Population structure

Principal component analysis was carried out on the additive genetic relationship matrix (base R 'prcomp') constructed from homozygous diallelic SNVs with no missing data (genotype set 2).

### Divergent regions

We thresholded divergent regions using kernel density smoothing (*Duong, 2020*) of the empirical distribution of $\theta_w$ across genomic windows (10 kb), taking the first positive value of the first derivative, after the minimum, as the threshold value. Regions were enumerated based simply on contiguous runs of the sign of the second derivative of $\theta_w$, that is, all local peaks in nucleotide diversity are treated as independent. This makes the unrealistic assumptions of uniform ancestral diversity and effective recombination, and is sensitive to sample size and binning. Deeper and broader population genetic data will be required to obtain more confident estimates of the number, size, and local structure of divergent regions, ideally with multiple high-quality genome assemblies to minimize confounding by reference mapping bias.

## Statistical analysis, data wrangling, and plotting

We used R (*R Development Core Team, 2018*) with packages data.table (*Dowle and Srinivasan, 2019*), dglm (*Dunn and Smyth, 2016*), dplyr (*Wickham, 2020*), ggmap (*Kahle and Wickham, 2013*), ggplot2 (*Wickham, 2016*), ggh4x (*van den Brand, 2020*), ggrepel (*Slowikowski, 2020*), lme4 (*Bates et al., 2015*), and tidyr (*Wickham and Henry, 2020*).

## Mating trials among isolates and RILs

Mating trials were initiated with one L4 hermaphrodite and one L4 male worm on a 6 cm NGM agarose plate seeded with 50 μL of OP-50 *E. coli*. Plates were scored 72 hr later, with success defined as the presence of multiple males in the $F_1$ generation.

We scored hermaphrodite cross probability in RILs by crossing NIC58 males to L4 RIL hermaphrodites. RIL trials ranged from 16 to 75 in number, with a median of 30, and took place over 116 days. A total of 338 JU1373 and 412 NIC58 control crosses were done on 107 of these days.

To estimate equilibrium male frequency, we scored sex ratio after 10 generations of passaging at large population size. Three L4 hermaphrodites and five L4 males were placed on a 6 cm agarose plate. Three days later, 3 mL of M9 buffer was pipetted onto the plate and 50 μL of worms was transferred to a 10 cm plate; 50 μL of worms was subsequently transferred to a 10 cm agarose plate every ~72 hr for 10 generations, at which point a sample of 267 ± 27 worms were sexed per strain. We performed three replicates of this passaging experiment.

Phenotypes for RIL QTL mapping were best linear unbiased predictions (BLUPs) from a binomial linear mixed-effects model (R package lme4; *Bates et al., 2015*). Mapping in R/qtl (*Broman et al., 2003*) used a 'normal' model, and 1000 permutations of the phenotype values to establish genome-wide significance. The variance explained by the single significant QTL was estimated from variance components by refitting the linear model to the raw data with a random effect of genotype within RIL.

## Genetic analysis of Medea activity

We used a standardized assay of the proportion of $F_2$ embryos that develop to adulthood according to wild-type schedule. $P_0$ males and hermaphrodites were paired as L4s, and the following day each hermaphrodite that bore a copulatory plug was transferred to a fresh plate to lay embryos. Two days later, when these embryos had developed to L4 stage, we isolated $F_1$ hermaphrodites overnight. The following day, hermaphrodites were singled to new plates and left to lay embryos for 8 hr. The hermaphrodites were then removed and the embryos on each plate counted. Three days later, when wild-type animals have reliably reached adulthood, we counted the adults on each plate by picking. In some cases, slow-developing animals that had reached L3 or L4 were counted separately. The majority of Medea-affected animals arrest as L1s and are difficult to see, so we typically estimated the number of arrested larvae as the number of embryos initially observed minus the number of adults counted 3 days later. In a small number of broods (~3%), the count of progeny at adulthood exceeded the count of embryos laid (by at most two extra adults, from broods containing ~35–55 total embryos laid). We made the assumption that this discrepancy reflected undercounting of embryos rather than overcounting of adults, given that embryos are hard to see. Thus, for such broods, we adjusted the embryo count upward to match the count of adults. All conclusions are robust to this adjustment.

Because of the low mating efficiency of many *C. tropicalis* genotypes, matings did not always produce cross-progeny. To distinguish self- and cross-progeny, we employed several approaches. In some experiments, we depleted hermaphrodites of sperm by transferring them to fresh plates on each of the first 4 days of adulthood, until they ceased reproduction. These sperm-depleted hermaphrodites could then be crossed to males, and resulting progeny inferred reliably to be cross offspring. This method is not suitable for all experiments because the sperm-depleted hermaphrodites have small broods and generally show age-associated decrepitude. As an alternative, we developed visible marker strains that allow us to distinguish self- and cross-progeny. We isolated a spontaneous Dumpy mutant in the JU1373 background and established strain QG2413, *Ctr-dpy* (*qg2*). Control experiments confirmed that this semidominant mutation allows for clean discrimination between Dpy and semi- or non-Dpy animals, and that the mutation is unlinked to the Medea loci on chromosomes III and V. Next, we generated a NIC58 derivative carrying a fluorescent reporter. Strain QG3501 (*qgIs5*) carries pCFJ104 [*Pmyo-3::mCherry::unc-54utr*] (*Frøkjaer-Jensen et al., 2008*). The transgene was introduced by microinjection into NIC58, integrated by UV, and backcrossed to NIC58 seven times. These animals express bright red fluorescence in muscle, visible under the dissecting scope from mid embryogenesis. Control experiments show that this transgene is unlinked to the Medea loci. To test for maternal-effect killing by the NIC58 haplotype on chromosome V, we generated strain QG4249, which carries the *qgIs5* transgene and the NIC58 mitochondrial genome and is homozygous NN at the chromosome III locus and JJ at the chromosome V locus. This strain was generated by crossing QG2514 males and QG3501 hermaphrodites and recovering a rare $F_2$ adult that was homozygous JJ at markers flanking the chromosome V transmission ratio distortion peak and also homozygous *qgIs5*.

Inheritance at Medea loci was tracked using PCR to amplify insertion/deletion markers near the transmission ratio distortion peaks:

> LG3.1336F TTAGAGCCGCTTGAAGTTGG
> LG3.1336R TCCGATGGACTAGGTTTCGT
> LG5.2017F TAACGCAATGGCCTCCTATC
> LG5.2017R GTTTGCTGGGTGGCCTAGTA

## Simulations

We used simulations to investigate the effects of selfing on the spread of a maternal-toxin/zygotic antidote haplotype in populations with the distinctive androdioecious mating system of *C. tropicalis*. Each simulated individual has a genotype and a sex. The single locus has haplotypes D and d. In single Medea simulations, D carries a maternal-effect toxin and zygotic-effect antidote, and d carries neither. In antagonistic Medea simulations, D and d carry two different Medea haplotypes. We initiate a population with $N$ individuals, and genotype frequencies and sex ratio that are at equilibrium given the population's (fixed) selfing rate $S$ and frequency $p$ of the haplotype D. The neutral

equilibrium inbreeding coefficient $F$ is $S/(2-S)$ and the male frequency is $(1-F)/2$. During the simulations $S$ is fixed but $F$ can be far from its equilibrium value due to selection.

For simulations in *Figure 10A and B*, with fixed population sizes, we generated starting populations using the equilibrium frequencies below.

Genotype Frequency
DD herm $((1-F)p^2 + pF)(1+F)/2$
Dd herm $(1-F)2 p(1\ p)(1+F)/2$
dd herm $((1-F)(1-p)^2 + (1-p)F)(1+F)/2$
DD male $((1-F)p^2 + pF)(1-F)/2$
Dd male $(1-F)2 p(1-p) (1-F)/2$
dd male $((1-F)(1-p)^2 + (1-p)F)(1-F)/2$

For simulations in *Figure 10C*, assessing the effects of patch dynamics on frequencies of antagonistic Medeas, we draw genotype frequencies from a multinomial according to the equilibrium frequencies, but we assign sexes deterministically to ensure every patch receives hermaphrodite founders.

We modeled the *C. tropicalis* androdioecious mating system, with self-fertile hermaphrodites that are incapable of mating with one another, and males that can cross-fertilize hermaphrodite oocytes. If no male is present in a population, each hermaphrodite produces a brood of size $B$ by selfing. If there are males, each hermaphrodite produces $SB$ hermaphrodite offspring by selfing and $(1-S)B$ offspring by mating (male or hermaphrodite with equal probability), with a single father drawn randomly from the population of males. Self-progeny are mostly hermaphrodites, except each has probability *Him* of being male (Him is the worm community name for the rate of male production by X-chromosome nondisjunction in selfing hermaphrodites).

For simplicity, we assume that all selection is on embryo viability and that there is no cost to the Medea allele. For single-Medea analyses (*Figure 10A and B*), individuals that are dd but have Dd mothers are viable with probability $V$ (i.e., if $V > 0$ some embryos can survive the maternal-effect toxin). Everybody else has viability 1.

|        | Offspring | | |
|--------|-----------|------|------|
| Mother | DD | Dd | dd |
| DD | 1 | 1 | - |
| Dd | 1 | 1 | $V$ |
| dd | - | 1 | 1 |

In simulations with antagonistic Medeas, we have two classes of affected offspring, with viabilities $V_1$ and $V_2$:

|        | Offspring | | |
|--------|-----------|------|------|
| Mother | DD | Dd | dd |
| DD | 1 | 1 | - |
| Dd | $V_1$ | 1 | $V_2$ |
| dd | - | 1 | 1 |

In simulations with fixed population size, each discrete generation is sampled from the pool of viable embryos. We then track $p$, the frequency of the D allele, given parameters $N$, $S$, $B$, $V$, and *Him*. In simulations with exponential growth, $p$ and $N$ are both variables. Simulations described in the text used $B = 50$, $V = V_1 = V_2 = 0.05$, and *Him* = 0.005, except for the case of unequal penetrances of antagonistic Medea elements, where we set $V_1$ and $V_2$ to 0.4 and 0.05 to model chromosome V. We then investigate the effects of population size ($N$) and selfing rate ($S$) on Medea allele frequency $p$. Simulations started with allele frequency of 0.05 for single-Medea scenarios, and with frequency 0.2 for antagonistic-Medea scenarios. Simulation code is available on github.

## Acknowledgements

We thank Arielle Martel, Jia Shen, and Patrick Ammerman for assistance in the lab, and the Félix, Teotónio, and Rockman labs for discussion. We are grateful to Marie-Anne Félix for the use of the JU strains, for stimulating discussions with her and Henrique Teotónio, and for helpful comments on the preprint from Asher Cutter. *C. tropicalis* strain QG843 was collected under permit SEX/A-25-12 from the Republic of Panama. Sequencing data were generated by the Duke University Center for Genomic and Computational Biology and the New York University Center for Genomics and Systems Biology Core Facility, and this work was supported in part through the NYU IT High Performance Computing resources, services, and staff expertise.

## Additional information

### Funding

| Funder | Grant reference number | Author |
|---|---|---|
| National Institute of Environmental Health Sciences | ES029930 | L Ryan Baugh<br>Erik C Andersen<br>Matthew V Rockman |
| National Institute of General Medical Sciences | GM089972 | Matthew V Rockman |
| National Institute of General Medical Sciences | GM121828 | Matthew V Rockman |
| New York University College of Arts and Sciences Dean's Undergraduate Research Fund | | John Yuen |
| Centre National de la Recherche Scientifique | | Christian Braendle |
| European Commission | H2020-MSCA-IF-2017–798083 | Luke M Noble |

The funders had no role in study design, data collection and interpretation, or the decision to submit the work for publication.

### Author contributions

Luke M Noble, Conceptualization, Data curation, Software, Formal analysis, Supervision, Funding acquisition, Investigation, Visualization, Methodology, Writing - original draft, Writing - review and editing; John Yuen, Data curation, Formal analysis, Funding acquisition, Investigation, Methodology; Lewis Stevens, Nicolas Moya, Data curation, Formal analysis; Riaad Persaud, Investigation, Writing - review and editing; Marc Moscatelli, Jacqueline L Jackson, Gaotian Zhang, Investigation; Rojin Chitrakar, Resources, Investigation; L Ryan Baugh, Resources, Supervision, Funding acquisition; Christian Braendle, Resources, Funding acquisition; Erik C Andersen, Resources, Supervision, Funding acquisition, Writing - review and editing; Hannah S Seidel, Conceptualization, Data curation, Formal analysis, Visualization, Methodology, Writing - original draft, Writing - review and editing; Matthew V Rockman, Conceptualization, Resources, Software, Formal analysis, Supervision, Funding acquisition, Investigation, Visualization, Methodology, Writing - original draft, Project administration, Writing - review and editing

### Author ORCIDs

Luke M Noble (ID) https://orcid.org/0000-0002-5161-4059
John Yuen (ID) http://orcid.org/0000-0002-1569-3298
Lewis Stevens (ID) http://orcid.org/0000-0002-6075-8273
Nicolas Moya (ID) https://orcid.org/0000-0002-6817-1784
Jacqueline L Jackson (ID) http://orcid.org/0000-0001-5376-0968
L Ryan Baugh (ID) http://orcid.org/0000-0003-2148-5492

Erik C Andersen [iD] http://orcid.org/0000-0003-0229-9651
Matthew V Rockman [iD] https://orcid.org/0000-0001-6492-8906

**Decision letter and Author response**
Decision letter https://doi.org/10.7554/eLife.62587.sa1
Author response https://doi.org/10.7554/eLife.62587.sa2

## Additional files

### Supplementary files

• Supplementary file 1. NIC58_rqtlCross.rda.zip; R/qtl cross object containing the NIC58 genetic map and associated recombinant inbred line (RIL) genotypes.

• Supplementary file 2. tropicalisGenomes.zip; archive containing nuclear and mitochondrial genomes and annotations for NIC58 and JU1373.

• Supplementary file 3. rawVariantCalls.zip; archive containing unfiltered variant calls for nuclear and mitochondrial genomes.

• Supplementary file 4. filteredVariantCalls.zip; archive containing hard-filtered variant calls for the nuclear genome.

• Supplementary file 5. processedVariantCalls.zip; archive containing hard-filtered variant calls for nuclear and mitochondrial genomes with no missing data.

• Supplementary file 6. caeno_orthogroups.tsv.zip; ortholog groupings for *Caenorhabditis* species.

• Transparent reporting form

### Data availability

All sequencing reads used in this project are available from the NCBI Sequence Read Archive under accession PRJNA662844. Software code is available from https://github.com/lukemn/tropicalis. (Copy archived at [https://archive.softwareheritage.org/swh:1:rev:6c45978ce4266975134e0aee0428-be4f4c5f2b6d].) All data generated or analysed during this study are included in the manuscript and supporting files. Source data and supplementary files have been provided for all figures.

The following dataset was generated:

| Author(s) | Year | Dataset title | Dataset URL | Database and Identifier |
|---|---|---|---|---|
| Noble LM, Yuen J, Chitrakar R, Baugh LR, Zhang G, Andersen EC, Rockman MV | 2020 | *C. tropicalis* genomic data | https://www.ncbi.nlm.nih.gov/bioproject/PRJNA662844 | NCBI BioProject, PRJNA662844 |

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
