## [Decision Letter]

**Acceptance summary:**

This impressive piece of work starts with the production of a chromosomal-scale genome assembly of the highly selfing and hitherto understudied species *Caenorhabditis tropicalis*. At the core of the study is the discovery and careful genetic dissection of a series of rarely characterized selfish elements causing maternally-inherited incompatibilities, called MEDEA elements. A simple population genetics model demonstrates that selfing effectively contributes to keep these selfish elements under check and prevents them from invading the genome.

**Decision letter after peer review:**

Thank you for submitting your article "Selfing is the safest sex for *Caenorhabditis tropicalis*" for consideration by *eLife*. Your article has been reviewed by three peer reviewers, including Vincent Castric as the Reviewing Editor and Reviewer #1, and the evaluation has been overseen by Detlef Weigel as the Senior Editor.

The reviewers have discussed the reviews with one another and the Reviewing Editor has drafted this decision to help you prepare a revised submission.

Summary:

The core of this beautiful study is the discovery and careful genetic dissection of maternal effect selfish elements causing zygote lethality in the understudied selfing species *Caenorhabditis tropicalis*. One of these selfish elements is characterized by antagonistic variants at a single genomic location. A simple simulation model explores the consequences of variations of the selfing rate on the establishment of these selfish elements, and focuses on the properties of the antagonistic variants. The paper also provides a wealth of genomic resources, including a well-assembled reference genome and a resequencing dataset revealing a poorly diverse genome punctuated by rare islands of higher divergence.

It has been reviewed by two reviewers and myself, and we all recognized a very important contribution to the field, both through the resources made available to establish *C. tropicalis* as a new model system, and through the precise genetic dissection of the incompatibilities that help think in a new way about the consequences of selfing.

Essential revisions:

After discussion with the reviewers, we agreed that the manuscript would in principle become acceptable for publication in *eLife* after the three following revisions :

1) First, the word "drive" is misleading in this context. Drive originally refers to meiotic distorters at the gametic level, while the maternally inherited incompatibilities reported here are acting at the zygotic level. This goes beyond a semantic issue, as the population genetic properties of such incompatibilities differ from classical drive elements (e.g. Wade and Beeman, 1994). As such, we feel that the manuscript will be confusing to many readers in its present form, preventing them to appreciate the true originality of the results. Replacing the term “drive” throughout the paper with something like “Maternal effect selfish elements” as in the original Wade and Beeman, 1994 paper, or any other more appropriate formulation is required to avoid the confusion.

2) Second, there was also a consensus that the genomic analysis is currently reported in a way that will distract the readers from the more interesting findings and proves particularly hard to follow. It is also poorly connected to the rest of the study. To improve readability, we ask you to drastically reduce this part to a short paragraph focusing on the key aspects that matter for the present study, and moving the detailed description of the results and comparison with other species to supplementary material.

3) Finally, the limitations of the simulations performed should be better acknowledged. They are an important part of the study, but they remain preliminary in the sense that the effect of variations of several key parameters is not explored. Limits to the scope and generality of the conclusions from the simulations should be better highlighted.

Reviewer #1:

This manuscript details a genetic and genomic analysis of the worm *Caenorhabditis tropicalis*, and proposes that its highly selfing mode of reproduction acts as an effective way to control the emergence of strong gene drive elements. The quality and amount of the work presented is impressive, regarding the power of the genomic analysis of structural variation, the level of details of the dissection of the genetic basis of the incompatibilities and the predictions derived from computer simulations of the fate of distorters along the outcrossing vs. selfing gradient. I have fairly substantial presentation issues I detail below to improve clarity.

1) The quantity of work in a single paper also means that the manuscript is currently very dense and quite hard to follow, with overly detailed descriptions of some results, a lack of precision in some others, and an overall lack of cohesion across the different parts of the work. For instance, how do the detailed analysis of correlates of genetic diversity along chromosomes across species precisely contribute to our understanding of the accumulation of distorters or mating system transitions? The genomic analysis is particularly lengthy and would be easier to follow if better focused on the aspects that are most relevant for the overall context of the study, or partly moved to supplementary material. In addition, several figures are poorly labeled, imprecisely called in the text (panels for supplementary figures are generally omitted), or even not called at all (see below for details comments and suggestions), making it extra hard to follow an already complicated flow of arguments.

2) The final claim of the paper is that segregation distorters have favoured the evolution of selfing, rather than being a consequence of it. Yet, the simulations do not actually address this issue. Instead, they focus exclusively on how well drivers can establish in populations with a fixed rate of selfing. The argument would be a lot more convincing if simulations actually followed the fate of mutants of the selfing rate once drivers have been introduced in the population, although this might easily be left for another study. At the very least the final suggestion should be toned down and formulated as a speculation. Also, it would be good to more clearly acknowledge the limits of these predictions. In particular, I suspect that many of them will depend on the strength of the incompatibilities, which were shown to vary in the genetic dissection but are considered fully lethal here. Also, the antagonistic drivers are assumed to have equal strength, and I suspect that their apparently neutral behaviour would not hold when they differ. Finally, a main result from the simulations is that selfers should be pretty good at eliminating simple drive systems, yet they seem to be quite common. How can this contradiction be solved ?

3) There is no experimental evidence for how the incompatibilities are determined at the mechanistic level, yet the paper varies in terminology and explicitly assumes a maternal toxin-zygotic antitoxin system at some places, especially in the simulation work. Sticking to a more generic formulation would be less confusing. It is also currently unclear whether the fact that the incompatibility loci are found within islets of genomic divergence is a significant observation, or just a fortuitous coincidence. Given the effort to define those divergent stretches, the authors seem to suggest that they played a disproportionate role, but this could be clarified.

Reviewer #2:

This paper investigates the genomic diversity and causes of outbreeding depression in a hitherto understudied selfing / androdioecious species of Caenorhabditis (C tropicalis). The main originality of the paper is the discovery and genetic dissection of maternally inherited incompatibilities (an incompatibility between mother and offspring due to a maternal toxin-zygotic antidote systems) that result in embryo/juvenile lethality in the progeny of heterozygotes, whereby one of the two homozygous genotypes is largely eliminated. The authors use a wealth of techniques and clever crosses to make this case, and argue that a predominantly selfing reproduction is an efficient way to reduce the cost of such incompatibilities, as it reduces the frequency of heterozygotes.

Overall this is a very well conducted study, with lots of details, an impressive amount of work (including, in passing, a thoroughly described high quality genome assembly for this species) and an exquisite care taken to check every hypothesis as best possible using crosses. I am quite convinced of the genetic incompatibility model which seem to have been tested in every thinkable way, and thus I have very little important concerns, none of which requires additional data, rather some rewording

1) The most important point is the vocabulary. The authors repeatedly use the word "drive" for what has been called maternally-inherited incompatibility or selfish element by their predecessors. It may seem purely conventional but it considerably obscures the reading -and most readers, that will (unfortunately) take the time for only a superficial reading will misunderstand what the authors mean… The word "drive" (meiotic drive or gene drive) usually describes meiotic distortion, i.e. nonmendelian proportions at the gamete stage. Here what we have is an incompatibility which results from an interaction between maternal and zygotic genotypes, and kills some of the offspring well after fertilization has taken place. It may sound boring, but replacing "drive" or "driver" everywhere in the manuscript by something like mother-offspring incompatibility or any other adequate term is absolutely necessary – otherwise the impact of this beautiful paper might be to generate confusion

2) A second, less important, comment is that the very last element of conclusion (that this kind of incompatibilities actually select for selfing) is the only bit that is not properly demonstrated. What the models document is actually the reverse (the impact of fixed selfing rates on incompatibilities); but none of these models actually involves modifiers of selfing rate. Sure, selfing reduces the population cost of incompatibilities, BUT a long history of models has made clear that there is no equivalence between this reduced cost and the fact that a modifier of selfer will easily hitchhike incompatibilities to invade a population. For example, it is not clear that outcrossing actually generates enough heterozygosity at incompatibility loci for this to occur in natural populations. So I'm happy with the assertion in the title (selfing is "safe") but not really with the idea that more selfing is selected when incompatibilities are present (while the reverse is proven, this kind of incompatibilities can reach higher frequency in the presence of selfing). This idea should be presented as a hypothesis… to be formally modelled and tested elsewhere.

Reviewer #3:

This article begins by describing the genome sequence of *C. tropicalis*, a self-fertile androdioecious species in the Caenorhabditis group. There have been at least 3 origins of self-fertility in Caenorhabditis and molecular and phylogenetic evidence indicate these reproductive mode transitions occurred independently. There are >10x as many described outcrossing dioecious species in the group and studying the 3 selfers is essential to understanding molecular evolution and reproductive mode transitions in this group. The selfers *C. elegans* and *C. briggsae* have nearly complete genome sequences for >20 and >15 years but initial efforts to sequence and study *C. tropicalis* were hobbled by contamination in the strain. Developing the *C. tropicalis* system is a huge contribution to the scientific community and this is an important piece of work.

This article does not stop at the genome sequence and covers a huge breadth. The authors sequence a number of strains to study population genetics, perform crosses and study loci consistent with genetic drive, and develop simulations to study the population dynamics of drive loci. The article is long and could really be 3 separate papers as the work presented in each section is detailed and interesting. Overall I find the work important, exciting and high-quality. However, there are several aspects of the article that require revision prior to publication.

1) The Abstract references gene drive elements that are consistent with maternal toxin/zygotic antidote systems but there is no mechanistic evidence presented that these are in fact maternal toxin/antidotes. Toxin/antidote systems like zeel/peel have been described in *C. elegans* (and recently in a preprint in *C. tropicalis*) but invoke a very specific mechanism where toxins are present in gametes and rescued by antidotes expressed in development. The authors need to scale this back, in some parts of the manuscript (for example the sections describing the crosses) they are very careful to refer to this as gene drive and in other parts (the simulations) it is speculative.

2) The length, breadth and complexity of the manuscript make it a bit difficult to get through. The supplementary material is embedded in the.pdf and I would not have been able to understand the paper without it. Unfortunately that makes it a full 48 pages of text that is not double-spaced, figures and tables. Some display items that are currently listed as supplementary, like Supplementary file 2 and Figure 7—figure supplement 1, should be moved to the main text and the lengthy description of the crosses should be slimmed down and/or partially moved to the supplementary materials.

3) The notation for the simulations was difficult to understand. It begins as a one locus model with haplotypes D and d but the second portion models two antagonistic drivers with the same notation. Similarly, the equilibrium inbreeding coefficient F^ is defined as *S*/(2-*S*) but the Frequency is given as 1-F^ below and I couldn't tell if F^ could evolve or not. I looked at the code on github and it was coded as (1-F^) but not clear if that was an initial condition.

4) Related to this, the simulations address a specific scenario of invasion by a gene drive element. The scope of the simulations is limited because of this scenario and some of the hard-coded parameters, for example a fixed cost to inbreeding and the brood proportions under selfing and outcrossing. One (highly speculative) explanation I've read for toxin/antidotes is that they originated as gamete competition proteins, possibly in multiple mating systems. Under this hypothesis the evolution of self-fertility could have been linked to an escape from toxin/antidotes as selfing makes them effectively neutral. In the Discussion the authors state that their simulations suggest antagonistic drive is unlikely to play a role in balancing selection but my intuition is that if certain parameters, for example viability and the inbreeding coefficient, were allowed to evolve the simulations would show different results. The authors do not need to develop extensive simulations but they do need to limit the scope of their discussion of their simulation results.

---

## [Author Response]

Reviewer #1:[…] I have fairly substantial presentation issues I detail below to improve clarity.1) The quantity of work in a single paper also means that the manuscript is currently very dense and quite hard to follow, with overly detailed descriptions of some results, a lack of precision in some others, and an overall lack of cohesion across the different parts of the work. For instance, how do the detailed analysis of correlates of genetic diversity along chromosomes across species precisely contribute to our understanding of the accumulation of distorters or mating system transitions? The genomic analysis is particularly lengthy and would be easier to follow if better focused on the aspects that are most relevant for the overall context of the study, or partly moved to supplementary material. In addition, several figures are poorly labeled, imprecisely called in the text (panels for supplementary figures are generally omitted), or even not called at all (see below for details comments and suggestions), making it extra hard to follow an already complicated flow of arguments.

We have now removed the genomic analysis entirely (which will form a short, new paper). As suggested, we retain only the minimum information required to investigate and interpret the incompatibilities – the genome assembly itself, and population genetics.

2) The final claim of the paper is that segregation distorters have favoured the evolution of selfing, rather than being a consequence of it. Yet, the simulations do not actually address this issue. Instead, they focus exclusively on how well drivers can establish in populations with a fixed rate of selfing. The argument would be a lot more convincing if simulations actually followed the fate of mutants of the selfing rate once drivers have been introduced in the population, although this might easily be left for another study. At the very least the final suggestion should be toned down and formulated as a speculation. Also, it would be good to more clearly acknowledge the limits of these predictions. In particular, I suspect that many of them will depend on the strength of the incompatibilities, which were shown to vary in the genetic dissection but are considered fully lethal here. Also, the antagonistic drivers are assumed to have equal strength, and I suspect that their apparently neutral behaviour would not hold when they differ.

We recognise that the scope of the simulations here is quite limited, and agree that a fuller exploration of the parameter space, including varying selfing rate, would be worthwhile in future work. In particular, the reviewer’s suggested modifier analysis is an important next step. We have now emphasised the restricted inferences that can be made from the simulated system, which was designed to test how variation in selfing rate controls the spread of Medea-type elements under simplified conditions.

The simulations in the original manuscript have penetrance fixed at 95%, and we share the reviewer’s expectation that unequal penetrances of antagonistic drivers would substantially change the evolutionary dynamics and outcomes. We have added a brief description of evolutionary dynamics when antagonistic drivers have unequal strength, similar to those we observed. In the discussion of simulation results, we now note:

“Effective neutrality in this scenario depends on the equal penetrance of antagonistic haplotypes in the model. […] Overall, selfing reduces Medea load both by decreasing heterozygote frequency and by inducing strong positive frequency-dependent selection that prevents antagonistic alleles from co-occurring.”

We are aware that any association between selfing rate and Medea elements is at this stage speculation, and have tried to be clear on this throughout the paper. We draw attention, for instance, to the final paragraph of the Introduction, in which “We *hypothesize* that offspring-killing haplotypes select for a high selfing rate in this species”, in the final sentence.

Finally, a main result from the simulations is that selfers should be pretty good at eliminating simple drive systems, yet they seem to be quite common. How can this contradiction be solved ?

This is a central question raised by our work. The simulations show that drive is not very effective under high rates of selfing, and so other processes may be at play, some of which we discuss throughout the Discussion (specifically in the subsections “Selfing and population genetics”, “Population dynamics of Medea elements” and “Strategies to combat Medea factors”. Our overall hypothesis is that Medeas can fix in local populations of selfers by drift (equally fit homozygous genotypes in highly selfing populations), but then they are not very good at spreading from population to population.

3) There is no experimental evidence for how the incompatibilities are determined at the mechanistic level, yet the paper varies in terminology and explicitly assumes a maternal toxin-zygotic antitoxin system at some places, especially in the simulation work. Sticking to a more generic formulation would be less confusing.

We have now adopted a more general terminology for these incompatibilities, following the maternal-effect precedent set by the Medea elements in *Tribolium castaneum*. Although the phenotypic consequences on offspring are not completely equivalent (effects here are mostly on larvae, not embryos), and the mechanistic basis of Medea factors is similarly obscure at present, we think this is an appropriate usage of this subclass of genetic drive. We continue to use “toxin” and “antidote,” with an express caveat, where we first introduce these terms, that they are shorthand for the genetic phenomena of “maternal-effect dominant larval arrest activity” and “zygotic-effect dominant rescue activity.”

It is also currently unclear whether the fact that the incompatibility loci are found within islets of genomic divergence is a significant observation, or just a fortuitous coincidence. Given the effort to define those divergent stretches, the authors seem to suggest that they played a disproportionate role, but this could be clarified.

We think the overlap of incompatibility and divergent loci is probably not coincidental.

More than 10 Mb of the *C. tropicalis* NIC58 genome is divergent in JU1373, however, and we have mapped three Medea loci to around 100 kb. So this is, at best, informed speculation at present. Our description is meant to provide a full account of data relevant to understanding the incompatibilities, and to connect our findings to precedents set by *C. elegans* Medeas and the *C. tropicalis* Medea characterised by Ben-David et al., 2020. As we note in the Discussion:

“The causal genes underlying the JU1373 and NIC58 drivers remain to be identified, but likely include one or more of the multiple genes unique to driver haplotypes, as seen in both *C. tropicalis* and *C. elegans* where toxin and antidote functions are encoded by genes present on the driver haplotype and absent (or pseudogenized) on the non-driver haplotype”.

The effort we have made to define these loci will guide future work on molecular and evolutionary mechanisms.

Reviewer #2:[…] 1) The most important point is the vocabulary. The authors repeatedly use the word "drive" for what has been called maternally-inherited incompatibility or selfish element by their predecessors. It may seem purely conventional but it considerably obscures the reading -and most readers, that will (unfortunately) take the time for only a superficial reading will misunderstand what the authors mean… The word "drive" (meiotic drive or gene drive) usually describes meiotic distortion, i.e. nonmendelian proportions at the gamete stage. Here what we have is an incompatibility which results from an interaction between maternal and zygotic genotypes, and kills some of the offspring well after fertilization has taken place. It may sound boring, but replacing "drive" or "driver" EVERYWHERE in the manuscript by something like mother-offspring incompatibility or any other adequate term is absolutely necessary – otherwise the impact of this beautiful paper might be to generate confusion.

We respect the reviewer’s strong views on the use of drive, and have replaced all specific usages with “Medea” elements. Our goal is effective communication, and we appreciate this guidance.

At the same time, we believe that current usage of “gene drive” is not at all restricted to the meiotic case. The most common usage at present is in the context of CRISPR-engineered gene drive, which is a non-meiotic mechanism, and the next most common is probably gamete-killing (e.g., *t*-alleles in mice and *sd* in flies), which acts postmeiotically. A recent major review of gene drive (Price et al., 2020), the result of a dedicated meeting on the topic, lists Medea-type elements in their Table 1, “Gene drive systems.” We favor a definition of drive that we employed in our 2008 paper (Seidel, Rockman and Kruglyak, 2008) about a paternal-effect Medea-like element: genic drive occurs when selection at the level of alleles within an individual (genic selection) acts independently of selection at the level of individuals within a population (genotypic selection).

We now note in the Introduction, “Haplotypes that behave in this way are known as Medea elements (Beeman et al., 1992; Beeman and Friesen, 1999), and they represent a form of post-zygotic gene drive (Price et al., 2020; Wade and Beeman, 1994).”

2) A second, less important, comment is that the very last element of conclusion (that this kind of incompatibilities actually select for selfing) is the only bit that is not properly demonstrated. What the models document is actually the reverse (the impact of fixed selfing rates on incompatibilities); but none of these models actually involves modifiers of selfing rate. Sure, selfing reduces the population cost of incompatibilities, BUT a long history of models has made clear that there is no equivalence between this reduced cost and the fact that a modifier of selfer will easily hitchhike incompatibilities to invade a population. For example, it is not clear that outcrossing actually generates enough heterozygosity at incompatibility loci for this to occur in natural populations. So I'm happy with the assertion in the title (selfing is "safe") but not really with the idea that more selfing is selected when incompatibilities are present (while the reverse is proven, this kind of incompatibilities can reach higher frequency in the presence of selfing). This idea should be presented as a hypothesis… to be formally modelled and tested elsewhere.

We agree completely, and aim to be clear in the manuscript that this is indeed a hypothesis, for instance, the final paragraph in the Introduction states, “We *hypothesize* that offspring-killing haplotypes select for a high selfing rate in this species”, and the very last sentence in the paper states, “incompatibility *may* also lead to increased selfing”, which we hope the reviewer can see are not strong assertions of fact. Indeed, we hope to explore this hypothesis through further, more detailed simulations.

Reviewer #3:[…] Overall I find the work important, exciting and high-quality. However, there are several aspects of the article that require revision prior to publication.1) The Abstract references gene drive elements that are consistent with maternal toxin/zygotic antidote systems but there is no mechanistic evidence presented that these are in fact maternal toxin/antidotes. Toxin/antidote systems like zeel/peel have been described in C. elegans (and recently in a preprint in C. tropicalis) but invoke a very specific mechanism where toxins are present in gametes and rescued by antidotes expressed in development. The authors need to scale this back, in some parts of the manuscript (for example the sections describing the crosses) they are very careful to refer to this as gene drive and in other parts (the simulations) it is speculative.

The reviewer is entirely correct that we do not yet have mechanistic (molecular) evidence that the Medea-type elements we have discovered are maternal toxins/zygotic antidotes. We have now added an introduction to Medea-type elements and we add the disclaimer that “we describe it as a toxin-antidote system for convenience, though the underlying mechanism of maternal-effect dominant lethality and zygotic-effect dominant rescue may be different.” This short-hand makes the models much easier to describe.

2) The length, breadth and complexity of the manuscript make it a bit difficult to get through. The supplementary material is embedded in the.pdf and I would not have been able to understand the paper without it. Unfortunately that makes it a full 48 pages of text that is not double-spaced, figures and tables. Some display items that are currently listed as supplementary, like Supplementary file 2 and Figure 7—figure supplement 1, should be moved to the main text and the lengthy description of the crosses should be slimmed down and/or partially moved to the supplementary materials.

We acknowledge this and have now greatly simplified the manuscript around the incompatibilities, moving most of the genomics and population genetics to a separate work. We will happily accept further advice about which supplementary figures should be in the main text.

3) The notation for the simulations was difficult to understand. It begins as a one locus model with haplotypes D and d but the second portion models two antagonistic drivers with the same notation. Similarly, the equilibrium inbreeding coefficient F^ is defined as S/(2-S) but the Frequency is given as (1-F^) below and I couldn't tell if F^ could evolve or not. I looked at the code on github and it was coded as (1-F^) but not clear if that was an initial condition.

We have rewritten the Materials and methods section to address these issues. We use the neutral equilibrium *Fhat* = *S*/(2-*S*) to determine the starting genotype frequencies for each simulation, but in subsequent generations only *S* is fixed and selection can drive *F* far from neutral equilibrium (as seen in Figure 10—figure supplement 2B).

4) Related to this, the simulations address a specific scenario of invasion by a gene drive element. The scope of the simulations is limited because of this scenario and some of the hard-coded parameters, for example a fixed cost to inbreeding and the brood proportions under selfing and outcrossing. One (highly speculative) explanation I've read for toxin/antidotes is that they originated as gamete competition proteins, possibly in multiple mating systems. Under this hypothesis the evolution of self-fertility could have been linked to an escape from toxin/antidotes as selfing makes them effectively neutral. In the Discussion the authors state that their simulations suggest antagonistic drive is unlikely to play a role in balancing selection but my intuition is that if certain parameters, for example viability and the inbreeding coefficient, were allowed to evolve the simulations would show different results. The authors do not need to develop extensive simulations but they do need to limit the scope of their discussion of their simulation results.

These are absolutely important caveats to our simulations, which address a much narrower question than our speculations in the Discussion. We have added text to make it clearer that the simulations do not allow the viabilities or selfing rates to evolve. We also add several sentences describing new simulations about the case of antagonistic Medea elements with unequal penetrances. Additions, as well as some relevant passages from the original version, are listed below.

Results (new): subsection “Medea dynamics in partial selfers”, first paragraph “Selfing rate is a fixed parameter in this system, and we leave a fuller exploration of the coevolutionary dynamics of selfing and intergenerational incompatibilities to future work”.

Results (new): subsection “Medea dynamics in partial selfers”, second paragraph “Effective neutrality in this scenario depends on the equal penetrance of antagonistic haplotypes in the model. […] Overall, selfing reduces Medea load both by decreasing heterozygote frequency and by inducing strong positive frequency-dependent selection that prevents antagonistic alleles from co-occurring”

Discussion (modified): subsection “Population dynamics of Medea elements”, second paragraph “antagonistic Medeas do not generate balancing selection, at least under the scenarios modeled”.

Materials and methods (modified): subsection “Simulations”, first paragraph “During the simulations *S* is fixed but *F* can be far from its equilibrium value due to selection.”

Materials and methods (modified): subsection “Simulations”, sixth paragraph.

Additional changes:

We performed new experiments to measure the maternal-effect killing of the NIC58 chromosome V haplotype. In our original submission, we had limited, imperfect data for this point, which was reported under the “NIC58 Driver” heading in Figure 9B. With a new strain that carries a fluorescent reporter to distinguish self- and cross-progeny, we were able to generate cleaner data, now reported in the same figure (now Figure 7B) under the “NIC58 Medea” heading. The qualitative results and interpretations are unchanged, but the sample sizes and quality of the data are improved.

The original text, “Crosses testing the JU1373 drivers showed that both acted via maternal, and not paternal, effect (Figure 9B). Crosses testing the NIC58 driver ruled out paternal effect (Figure 9B); crosses testing a maternal effect were confounded by the poor mating ability of JU1373 males, but were consistent with a maternal effect (Figure 9B). We therefore infer a maternal effect for the NIC58 driver, and conclude that all three driver alleles act via maternal, and not paternal, effect (Figure 9C),” now reads “Crosses testing the JU1373 and NIC58 Medeas showed that all acted via maternal, and not paternal, effect (Figure 7B).” We also added the details of the new cross to the Materials and methods section.